# Gated Relational Alignment via Confidence-based Distillation for Efficient VLMs

**Yanlong Chen** [1 2]  **Amirhossein Habibian** [3]  **Luca Benini** [1 4]  **Yawei Li** [1 2]

## Abstract

Vision-Language Models (VLMs) deliver strong multimodal performance, but are costly to deploy, with post-training quantization often causing significant accuracy degradation. Despite its potential, quantization-aware training (QAT) for VLMs remains underexplored. We propose GRACE, a framework that unifies knowledge distillation and QAT under the Information Bottleneck principle, where quantization constrains information capacity and distillation determines what to preserve. To achieve optimal distillation, we adopt a student-teacher formulation, where the teacher is treated as a proxy for task-relevant information. We introduce confidence-gated decoupled distillation to filter unreliable supervision, relational-centered kernel alignment to transfer visual token structures, and an adaptive controller to balance fidelity with capacity constraints. Extensive evaluations of the LLaVA and Qwen benchmarks show that our INT4 models consistently outperform the BF16 baselines (e.g., LLaVA-1.5-7B: 70.1 vs. 66.8 on SQA; Qwen2-VL-2B: 76.9 vs. 72.6 on MMBench), closely matching teacher performance. With real INT4 kernels, we achieve $3\times$ throughput and 54% memory reduction. Code and data are available at: ForeverBlue816/GRACE.

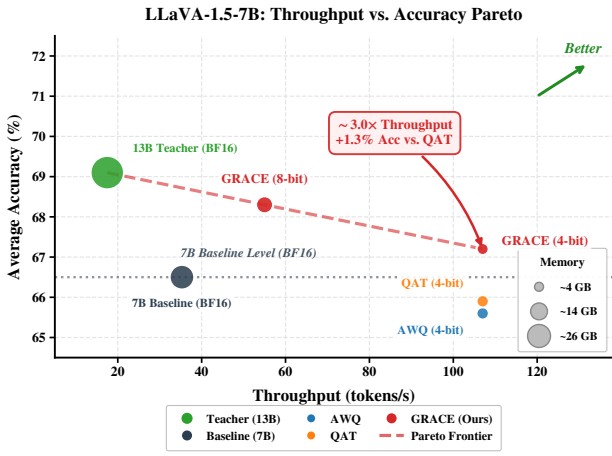

*Figure 1.* Throughput-accuracy Pareto analysis on LLaVA-1.5-7B. Bubble size indicates GPU memory footprint. **GRACE** achieves higher throughput than BF16 baselines while maintaining stronger accuracy than existing quantization methods.

## 1. Introduction

Vision-language models (VLMs) have become a central paradigm for multimodal intelligence, demonstrating strong capabilities across visual question answering, complex scene understanding, image-text reasoning, vision-language-

action systems, and broader multimodal visual applications (Bai et al., 2023; Liu et al., 2023; 2024a; Kawaharazuka et al., 2025; Liang et al., 2025). However, these capabilities come with substantial computational cost. Modern VLMs typically contain billions of parameters and require large memory bandwidth, making them difficult to deploy on resource-constrained platforms. This motivates low-bit compression methods that can reduce inference cost while preserving the reasoning and perception ability of full-precision models.

Quantization is one of the most effective tools for efficient deployment. Post-training quantization (PTQ) is attractive because it requires little or no retraining (Frantar et al., 2022; Shao et al., 2023; Lin et al., 2024), but aggressive INT4 quantization often causes severe degradation in VLMs due to heterogeneous multimodal distributions and cross-modal sensitivity (Wang et al., 2024a; Li et al., 2025). Quantization-aware training (QAT) provides a stronger alternative by exposing the model to low-precision constraints during optimization. Recent studies in language and vision models suggest that accurate quantization requires not only preserving task performance but also explicitly managing quantization-induced errors (Liu et al., 2024d; 2025; Tian et al., 2025). Nevertheless, QAT for VLMs remains underexplored, since VLMs combine visual representations, mul-

---

[1]Department of Information Technology and Electrical Engineering, ETH Zurich, Zurich, Switzerland [2]School of Electrical and Electronic Engineering, Nanyang Technological University, Singapore [3]Qualcomm AI Research, Amsterdam, the Netherlands [4]Department of Electrical, Electronic and Information Engineering, University of Bologna, Bologna, Italy. Correspondence to: Yawei Li <li.yawei.ai@gmail.com>.

*Proceedings of the 43rd International Conference on Machine Learning*, Seoul, South Korea. PMLR 306, 2026. Copyright 2026 by the author(s).

timodal projection, and language decoding, each with different sensitivity to low-bit perturbations (Sun et al., 2024; Jin et al., 2025).

Meanwhile, knowledge distillation has emerged as a powerful tool for compressing VLMs (Cao et al., 2025; Lee et al., 2025). A natural idea is to combine KD with QAT, using a strong teacher to guide the low-bit student. However, this combination is non-trivial. First, teacher predictions are not uniformly reliable: high-entropy or biased predictions may introduce noisy supervision. Second, a quantized student has limited representational capacity and cannot faithfully absorb all teacher information. This issue is especially important for VLMs, where predictions can be affected by prior preferences, class-level confusion, and prompt-induced uncertainty (Tian et al., 2026). Therefore, effective QAT-based VLM compression requires a mechanism that identifies which teacher information is worth preserving under a strict bit budget.

We interpret this problem through the Information Bottleneck (IB) principle (Tishby et al., 2000; Tishby & Zaslavsky, 2015). Quantization imposes a hard capacity constraint by reducing representational precision, while distillation provides dense supervision about task-relevant information. From this perspective, VLM compression becomes a capacity allocation problem: the quantized student must retain high-value teacher knowledge while discarding noisy or redundant information. Standard QAT relies mainly on task losses, which are often too sparse to guide this allocation, especially for multimodal tasks where supervision is distributed across visual tokens, text tokens, and answer logits. We therefore use the teacher as a proxy for task relevance and dynamically regulate distillation according to confidence and information preservation.

Building upon this insight, we propose **GRACE** (**G**ated **R**elational **A**lignment via **C**onfidence-based Distillation for **E**fficient VLMs), a unified framework for QAT-based VLM compression. GRACE consists of three complementary components: (i) *confidence-gated decoupled knowledge distillation*, which suppresses noisy supervision from uncertain teacher predictions; (ii) *relational centered kernel alignment*, which transfers visual-token relational structure rather than point-wise features; and (iii) an *adaptive information-bottleneck controller*, which dynamically balances teacher guidance with the student's quantized capacity. For low-bit training, GRACE further employs group-wise learned step-size quantization.

Extensive experiments on LLaVA-1.5 and Qwen2-VL demonstrate the effectiveness of GRACE. Our distilled LLaVA-1.5-7B achieves 69.0% average accuracy, improving the 7B baseline by **3.8%** and nearly matching the 13B teacher. More importantly, as shown in Figure 1, our INT4 Qwen2-VL-2B not only recovers but surpasses the full-

precision baseline across all benchmarks, such as **79.1 vs. 73.7** on ScienceQA. GRACE also delivers significant inference speedup and memory reduction when deployed with real INT4 kernels.

Our contributions are summarized as follows:

- We formulate QAT-based VLM compression from an Information Bottleneck perspective, connecting low-bit quantization, capacity allocation, and teacher-guided knowledge preservation.

- We propose GRACE, a unified framework that combines confidence-gated distillation, relational CKA, adaptive IB control, and group-wise learned step-size quantization for efficient VLM compression.

- We show that GRACE enables INT4-quantized VLMs to surpass BF16 baselines while achieving substantial inference speedup and memory reduction.

**Conflict of Interest Disclosure.** All authors are affiliated with academic institutions (ETH Zurich and University of Bologna) and declare no financial conflicts of interest; the models evaluated in this work (LLaVA-1.5 and Qwen2-VL) are publicly released open-source models.

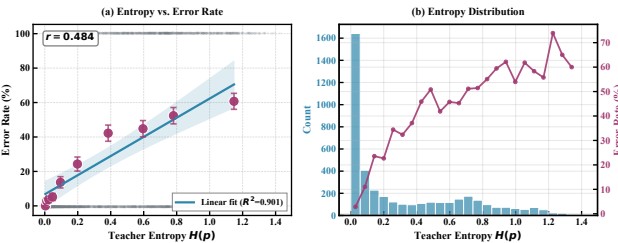

*Figure 2.* Correlation between teacher entropy and error rate on ScienceQA (LLaVA-1.5 13B). Higher entropy consistently corresponds to higher prediction error, supporting entropy as a confidence signal for gated distillation.

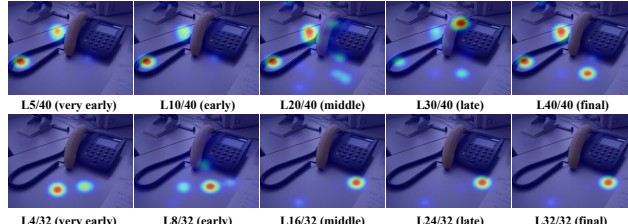

*Figure 3.* Multi-layer attention visualization of LLaVA-1.5 13B (top) and 7B (bottom). Given the question "What object is being used as the telephone receiver?", the 13B model progressively localizes the banana, while the 7B model shows scattered attention.

## 2. Motivation

Before presenting our method, we conduct empirical analyzes to identify key challenges in distilling VLMs.

## 2.1. Teacher Confidence and Supervision Quality

Standard knowledge distillation treats all teacher predictions equally, implicitly assuming uniform supervision quality across tokens. However, we hypothesize that teacher predictions vary significantly in reliability: uncertain outputs may introduce noise rather than valuable knowledge. To test this, we examine the relationship between teacher confidence and prediction accuracy in the ScienceQA data set using LLaVA-1.5 13B as the teacher. We measure uncertainty by the entropy of the output distribution $H(P_T) = -\sum_v P_T(v) \log P_T(v)$. As shown in Figure 2, we observe a significant positive correlation between teacher entropy and error rate at the sample level (Pearson $r = 0.484$). When aggregating samples into deciles of entropy, linear regression on binned error rates yields $R^2 = 0.901$, which confirms a strong monotonic relationship. This empirical observation is consistent with Fano's inequality, which theoretically links higher entropy to an elevated lower bound on error probability (Verdú et al., 1994), motivating our **confidence-gated distillation** mechanism (Section 3.2).

## 2.2. Visual Attention Mismatch Across Model Scales

Although logit-based distillation transfers output predictions, larger teacher models may possess superior visual reasoning capabilities not captured at the output level. To investigate, we visualize attention patterns of LLaVA-1.5 13B and 7B models across layer depths. Given an image where a banana is used as a telephone receiver, we ask "What object is being used as the telephone receiver?"

As shown in Figure 3, a striking mismatch emerges: the teacher progressively refines its attention, successfully localizing the banana in later layers. In contrast, the student displays scattered attention across all layers, failing to identify the region relevant to the task. This reveals that the teacher's superior visual reasoning comes from its ability to develop semantically meaningful attention patterns, a capability that logit-based distillation alone cannot transfer, motivating our **Relational Centered Kernel Alignment** loss (Section 3.3).

## 3. Method

### 3.1. Overview

Quantization restricts the capacity of the model to a fixed bit budget, forcing the network to prioritize which information to retain. We present GRACE, a framework that formulates this capacity allocation problem through the lens of the Information Bottleneck principle: the quantized student must retain task-relevant knowledge from the teacher while discarding redundant information that cannot be faithfully represented under bit-width constraints. As illustrated in Figure 4, a frozen teacher and a trained quantized student process the same input to produce output distributions $P_T$ and $P_S$, respectively. The student is trained using a combination of cross-entropy loss, confidence-gated decoupled distillation loss, and relational alignment loss, with the distillation weight dynamically adjusted by an adaptive IB controller. We provide a detailed description of each component in the following.

### 3.2. Confidence-Gated Decoupled Knowledge Distillation

Standard knowledge distillation (Hinton et al., 2015) minimizes the KL divergence between teacher and student output distributions. However, this approach treats all teacher predictions equally, ignoring the varying reliability of teacher supervision across different tokens. We address this limitation through two mechanisms: decoupled knowledge distillation and confidence-based gating.

**Decoupled Knowledge Distillation.** Following (Zhao et al., 2022), we decompose the distillation loss into two components that capture distinct aspects of the teacher's knowledge. Let $P_T$ and $P_S$ denote the probability distributions of the teacher and student on the vocabulary, and let $y$ be the ground-truth label. We define the following two components of the distillation loss:

- **Target Class Knowledge Distillation (TCKD):** This component captures the teacher's confidence in the correct answer by comparing binary distributions over the target versus non-target classes:

$$\mathcal{L}_{\text{TCKD}} = D_{\text{KL}} \left( [P_T^t, 1 - P_T^t] \,\|\, [P_S^t, 1 - P_S^t] \right) \quad (1)$$

where $P_T^t = P_T(y)$ and $P_S^t = P_S(y)$ are the probabilities assigned to the target class.

- **Non-target Class Knowledge Distillation (NCKD):** This term transfers the "dark knowledge" embedded in the teacher's distribution over incorrect classes:

$$\mathcal{L}_{\text{NCKD}} = D_{\text{KL}} \left( \hat{P}_T^{\text{nt}} \,\|\, \hat{P}_S^{\text{nt}} \right) \quad (2)$$

where $\hat{P}^{\text{nt}}$ denotes the renormalized distribution over non-target classes.

The per-token *Decoupled Knowledge Distillation* (DKD) loss combines these components:

$$\mathcal{L}_{\text{DKD}}^{(i)} = \alpha \cdot \mathcal{L}_{\text{TCKD}}^{(i)} + \beta_{\text{dkd}} \cdot \mathcal{L}_{\text{NCKD}}^{(i)} \quad (3)$$

where $i$ indexes tokens and $\alpha$, $\beta_{\text{dkd}}$ are weighting coefficients. TCKD captures the teacher's confidence in the ground-truth token, while NCKD transfers the relational structure among all other tokens in the vocabulary. Following (Zhao et al., 2022), we set $\beta_{\text{dkd}} > \alpha$ to emphasize the rich "dark knowledge" encoded in non-target distributions,

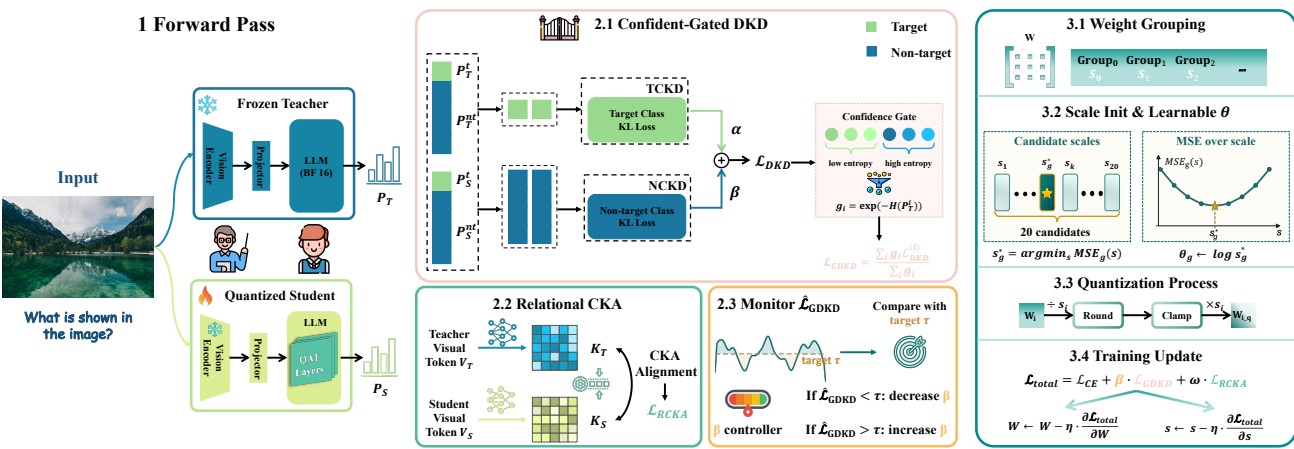

*Figure 4.* Overview of the GRACE framework. A frozen teacher and a quantization-aware student jointly process each input. The student receives three complementary supervisory signals: (i) **Confidence-Gated DKD**, which decomposes distillation into target-class and non-target-class components and weights token-level supervision by teacher confidence; (ii) **Relational CKA**, which aligns centered kernel matrices $K_T$ and $K_S$ of visual tokens at the penultimate LLM layer while excluding text tokens; and (iii) an **Adaptive IB Controller**, which monitors the EMA-smoothed gated distillation loss $\widehat{\mathcal{L}}_{\text{GDKD}}$ and adaptively adjusts the distillation strength $\beta$. For QAT, the student weights are partitioned into groups with independent learnable scales. Each scale is initialized by MSE-based grid search, parameterized as a log-scale $\theta_g = \log s_g$, and used in round-and-clamp fake quantization. During training, both weights $W$ and quantization scales $s$ are jointly optimized under $\mathcal{L}_{\text{total}} = \mathcal{L}_{\text{CE}} + \beta \cdot \mathcal{L}_{\text{GDKD}} + \omega \cdot \mathcal{L}_{\text{RCKA}}$.

which has been shown to be more informative for knowledge transfer than target-class probabilities alone.

**Confidence-Based Gating.** As demonstrated in Section 2.1, teacher entropy exhibits a strong correlation with prediction errors, indicating that high-entropy outputs constitute unreliable supervision signals. Motivated by this observation, we introduce a confidence-based gating mechanism that adaptively modulates the distillation loss according to teacher certainty.

For each token $i$, we compute the entropy of the teacher's output distribution:

$$H_i = H(P_T^{(i)}) = -\sum_v P_T^{(i)}(v) \log P_T^{(i)}(v) \quad (4)$$

We normalize this entropy as $\tilde{h}_i = H_i / \log|V| \in [0, 1]$, where $|V|$ denotes the vocabulary size. The confidence weight for token $i$ is then defined as:

$$g_i = \exp\left(-\tilde{h}_i\right) \quad (5)$$

This exponential formulation assigns high weights to confident teacher predictions (low entropy) while suppressing noisy supervision from uncertain predictions (high entropy). The gated DKD loss aggregates per-token losses with confidence weighting:

$$\mathcal{L}_{\text{GDKD}} = \frac{\sum_i g_i \cdot \mathcal{L}_{\text{DKD}}^{(i)}}{\sum_i g_i} \quad (6)$$

where the summation is over all valid tokens in the batch.

**Information-Theoretic Justification.** The gated distillation loss admits a principled interpretation under the Information Bottleneck framework. By defining the importance-reweighted distribution:

$$p_g(i) \triangleq \frac{g_i}{\sum_j g_j} \quad (7)$$

we can express $\mathcal{L}_{\text{GDKD}} = \mathbb{E}_{p_g}[\mathcal{L}_{\text{DKD}}^{(i)}]$. This reveals that confidence gating implements a *bottleneck on the supervision signal*, allocating distillation capacity towards tokens where the teacher posterior is sharp (low entropy), i.e., where the teacher provides higher information content.

**Theorem 3.1** (Effect of Confidence Gating). *Let $w_i = g_i / \sum_j g_j$ denote the normalized weights. The gated loss satisfies the following:*

$$\mathcal{L}_{GDKD} = \bar{\mathcal{L}}_{DKD} + N \cdot \text{Cov}\left(w_i, \mathcal{L}_{DKD}^{(i)}\right) \quad (8)$$

*where $\bar{\mathcal{L}}_{DKD} = \frac{1}{N}\sum_i \mathcal{L}_{DKD}^{(i)}$ is the unweighted average. Since $w_i$ decreases monotonically with $\tilde{h}_i$, positive correlation between entropy and loss implies $\text{Cov}(w_i, \mathcal{L}_{DKD}^{(i)}) < 0$.*

The proof is provided in Appendix B.2.

**KL Divergence as Information Gap.** Beyond filtering unreliable supervision, the KL-based distillation objective itself admits an information-theoretic interpretation. Let $Y_T$ denote a pseudo-label sampled from the teacher distribution, i.e., $Y_T \mid X = x \sim P_T(\cdot \mid x)$, and let $Z_S = f_S(X)$ represent the student's learned representation. Using the student's output $P_S$ as a variational decoder yields:

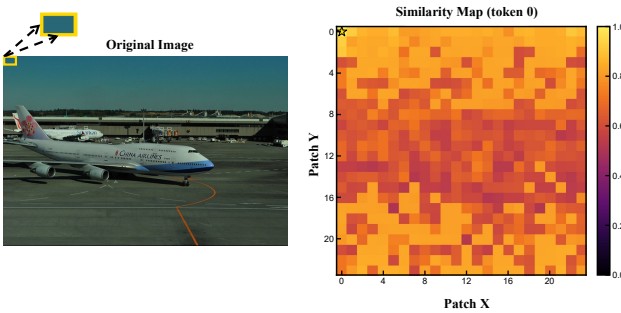

*Figure 5.* Visualization of pairwise visual token similarity from LLaVA-1.5 13B. The heatmap shows cosine similarity between token 0 (yellow box, located in the sky region) and all other tokens in the spatial grid, where Patch X and Patch Y denote the horizontal and vertical grid coordinates respectively.

**Proposition 3.2** (Variational Lower Bound and KL Gap).

$$I(Z_S; Y_T) \geq I(X; Y_T) - \mathbb{E}\left[D_{KL}\left(P_T \| P_S\right)\right] \quad (9)$$

The proof is provided in Appendix B.3.

This result provides a clean interpretation: the KL divergence quantifies the *information gap* between the maximum teacher information $I(X; Y_T)$ and the information captured by the student $I(Z_S; Y_T)$. Consequently, minimizing $\mathcal{L}_{\text{GDKD}}$ directly maximizes the mutual information between the student's representation and the teacher's knowledge.

### 3.3. Relational Centered Kernel Alignment

As demonstrated in Section 2.2, larger teacher models develop superior visual attention patterns that logit-level distillation alone cannot transfer. While relational knowledge distillation methods (Park et al., 2019) have shown that transferring inter-sample structural relationships improves student learning, these approaches operate at the batch level and fail to capture the fine-grained *intra-sample* relational structures among visual tokens that are critical for visual reasoning. We propose Relational Centered Kernel Alignment (RCKA) to explicitly transfer this token-level relational knowledge to the student.

**Relational Structure in Visual Representations.** We first investigate the internal structure of teacher representations. As illustrated in Figure 5, we visualize pairwise similarities between visual tokens from the 13B teacher. By selecting a token from the sky region (yellow box) and computing its similarity with all other tokens, we observe that the sky token exhibits high similarity with other sky tokens while showing minimal similarity with the aircraft or ground regions. This demonstrates that the teacher's intermediate hidden states from the language model backbone encode rich relational structures, organizing visual tokens into semantically coherent regions.

**RCKA for Relational Knowledge Transfer.** Motivated

by these observations, we propose **Relational Centered Kernel Alignment (RCKA)** to transfer the teacher's relational visual knowledge to the student. While CKA (Kornblith et al., 2019) has been explored for knowledge distillation (Saha et al., 2022; Chen et al., 2025), our approach differs in several key aspects: (i) existing methods perform *layer-wise* alignment between corresponding layers, whereas RCKA targets *intra-sample* relational structures among visual tokens; (ii) prior relational KD methods like RKD (Park et al., 2019) capture *inter-sample* relationships at the batch level, while RCKA preserves fine-grained pairwise similarities that organize visual tokens into semantically coherent regions (e.g., sky tokens clustering together, as shown in Figure 5); and (iii) RCKA specifically addresses VLMs, where visual tokens processed through the LLM backbone develop rich relational structures critical for visual reasoning, which logit-level distillation cannot transfer.

Specifically, let $V_T \in \mathbb{R}^{n \times d_T}$ and $V_S \in \mathbb{R}^{n \times d_S}$ denote the visual token representations extracted from the penultimate layers of the teacher and student LLM backbones, respectively, where $n$ is the number of visual tokens. Text tokens are excluded entirely from the Gram matrix computation to ensure RCKA transfers visual perception capabilities without conflating visual and linguistic representations. We select the penultimate layer because prior work (Kornblith et al., 2019; Chen et al., 2025) has demonstrated that intermediate layers capture richer semantic features than the final layer, which tends to be task-specific.

**Gram Matrix Computation.** We compute normalized Gram matrices that capture pairwise similarities among visual tokens:

$$K_T = \bar{V}_T \bar{V}_T^\top, \quad K_S = \bar{V}_S \bar{V}_S^\top \quad (10)$$

where $\bar{V} = V/\|V\|_2$ denotes row-wise $\ell_2$ normalization. Each entry $(K)_{ij}$ measures the cosine similarity between visual tokens $i$ and $j$.

**Centered Kernel Alignment.** To ensure invariance to the mean of representations, we apply centering to the Gram matrices:

$$\tilde{K} = HKH, \quad \text{where} \quad H = I_n - \frac{1}{n}\mathbf{1}_n\mathbf{1}_n^\top \quad (11)$$

Here, $I_n$ is the identity matrix and $\mathbf{1}_n$ is the all-ones vector.

To compare the relational structures encoded in $\tilde{K}_T$ and $\tilde{K}_S$, we adopt Centered Kernel Alignment (CKA) (Kornblith et al., 2019), which builds upon the Hilbert-Schmidt Independence Criterion (HSIC). HSIC is a kernel-based statistical measure of dependence between two sets of variables: given centered kernel matrices, it computes their normalized inner product in the space of Hilbert-Schmidt operators as $\text{HSIC}(K_T, K_S) = \frac{1}{(n-1)^2}\text{Tr}(\tilde{K}_T\tilde{K}_S)$. Intuitively, HSIC quantifies how well the pairwise relationships

in one representation predict those in another. If visual tokens that are similar in the teacher's representation are also similar in the student's, HSIC will be high.

CKA normalizes HSIC to obtain a similarity measure bounded in $[0, 1]$:

$$\text{CKA}(K_T, K_S) = \frac{\text{HSIC}(K_T, K_S)}{\sqrt{\text{HSIC}(K_T, K_T) \cdot \text{HSIC}(K_S, K_S)}} \tag{12}$$

This normalization is crucial for our setting: it renders CKA invariant to isotropic scaling of representations and, importantly, enables meaningful comparison even when the teacher and student have different hidden dimensions $(d_T \neq d_S)$. Unlike point-wise alignment methods such as MSE that require matching dimensions, CKA operates on $n \times n$ Gram matrices and thus bypasses the need for projection layers.

The RCKA loss encourages the student to preserve the teacher's relational structure:

$$\mathcal{L}_{\text{RCKA}} = 1 - \text{CKA}(K_T, K_S) \tag{13}$$

By aligning relational structures rather than point-wise features, RCKA facilitates knowledge transfer even when $d_T \neq d_S$, enabling the student to acquire the geometric properties underlying the teacher's superior visual perception and progressive attention refinement.

### 3.4. Group-wise Learned Step-size Quantization

To enable efficient low-bit inference, we employ QAT with learned step sizes (Esser et al., 2019). Unlike PTQ methods that fix quantization scales after calibration, our method treats the scales as learnable parameters and optimizes them jointly with model weights under the distillation objective.

**Group-wise Quantization.** VLM weights exhibit heterogeneous distributions across layers and channels due to their multimodal structure (Wang et al., 2024a; Guo et al., 2025). Per-tensor quantization is often too coarse to capture such variation, while channel-wise quantization still ignores fine-grained differences within each channel. Inspired by microscaling formats (Rouhani et al., 2023), we adopt group-wise quantization, which partitions weights into contiguous groups and assigns each group an independent scale.

For a weight matrix $W$, we flatten it and partition it into $G = d_{\text{out}} d_{\text{in}}/g$ groups, denoted as $W \rightarrow \{W_i\}_{i=0}^{G-1}$ with $W_i \in \mathbb{R}^g$. We use $g = 128$ by default. Each group has a learnable log-scale $\theta_i$ with $s_i = \exp(\theta_i)$, which guarantees positive scales and stabilizes multiplicative updates. Quantization is applied only to the LLM decoder linear layers, while the vision-side and output-related modules remain in BF16 to preserve visual encoding and output distribution quality.

**Learned Step-size Quantization.** Following LSQ (Esser

et al., 2019), we parameterize scales in log space and initialize each group scale by minimizing fake-quantization reconstruction error. For group $W_i$, we sample $K = 20$ candidate scales from $\mathcal{S}_i = [0.3, 1.2] \cdot \max(|W_i|)/Q_p$. We define

$$q_i(s) = \text{clip}\left(\lfloor W_i/s \rceil, -Q_n, Q_p\right),$$
$$s_i^{(0)} = \arg\min_{s \in \mathcal{S}_i} \|W_i - sq_i(s)\|_2^2. \tag{14}$$

We set $\theta_i^{(0)} = \log s_i^{(0)}$. This MSE-based initialization is particularly important for 4-bit QAT, where the quantization grid is coarse and max-based scales can leave large rounding or clipping errors. During training, each group is fake-quantized as $\widehat{W}_i = s_i q_i(s_i)$, where $\lfloor \cdot \rceil$ denotes round-to-nearest. We use the straight-through estimator for rounding, allowing both $W$ and $\theta$ to receive gradients from the downstream distillation objectives.

**Quantization as an Information Capacity Constraint.** The Information Bottleneck principle (Tishby et al., 2000) seeks to preserve task-relevant information while limiting representation complexity. In our setting, low-bit quantization naturally imposes such a constraint by reducing each weight from 16-bit precision to $b$-bit precision. We therefore view quantized distillation as preserving teacher-relevant information under a hard capacity limit:

$$\max I(Z_S; Y_T) \quad \text{s.t.} \quad I(Z_S; X) \leq C_b. \tag{15}$$

Here, $Z_S$ denotes the student representation and $Y_T$ denotes teacher-provided task-relevant information. Since the quantizer physically enforces the capacity limit $C_b$, we do not add a separate penalty on $I(Z_S; X)$. Instead, we require the quantized student to retain sufficient teacher knowledge by constraining the distillation loss, written as $\min \mathcal{L}_{\text{task}}$ subject to $\mathcal{L}_{\text{distill}} \leq \tau$. The corresponding Lagrangian is

$$\mathcal{L}_{\text{task}} + \beta \left(\mathcal{L}_{\text{distill}} - \tau\right), \tag{16}$$

where $\beta$ is adapted online through projected dual ascent on the smoothed distillation loss, as described in Sec. 3.2. This formulation connects quantization and distillation: quantization limits the student's information capacity, while distillation guides how this limited capacity should be allocated.

**Joint Optimization.** The model weights $W$ and log-scales $\theta$ are jointly optimized with $\mathcal{L}_{\text{total}} = \mathcal{L}_{\text{CE}} + \beta \mathcal{L}_{\text{GDKD}} + \omega \mathcal{L}_{\text{RCKA}}$. Their updates are

$$W \leftarrow W - \eta_W \nabla_W \mathcal{L}_{\text{total}}, \qquad \theta \leftarrow \theta - \eta_\theta \nabla_\theta \mathcal{L}_{\text{total}}. \tag{17}$$

We place $\theta$ in a separate optimizer group with no weight decay and use $\eta_\theta = 10\eta_W$. This larger learning rate helps the scales quickly track the changing weight distribution during QAT, which is especially important in early training. Empirically, this follows standard LSQ practice (Esser et al., 2019) and improves 4-bit stability. The joint update allows weights to migrate toward the quantization grid while the scales co-evolve to preserve task-relevant information.

*Table 1.* Comparison with knowledge distillation methods on LLaVA-1.5. All student models use Vicuna-7B as the language backbone. Best results among 7B models are **bolded**, second best are underlined.

| Method | VQA$^{v2}$ | GQA | TextVQA | VizWiz | POPE | SQA | MME | MMB | Avg. |
|---|---|---|---|---|---|---|---|---|---|
| LLaVA-1.5-13B (Teacher) | 80.0 | 63.5 | 61.3 | 53.6 | 85.9 | 71.6 | 1531.3 | 67.7 | 69.1 |
| LLaVA-1.5-7B (Baseline) | 78.5 | 62.0 | 58.2 | 50.0 | 85.9 | 66.8 | 1510.7 | 64.3 | 66.5 |
| MoVE-KD-v1.0 [CVPR'25] | 79.5 | 63.2 | 58.3 | 52.3 | 86.9 | 69.3 | 1524.5 | 66.3 | 68.0 ↑1.5% |
| MoVE-KD-v1.1 [CVPR'25] | **79.9** | **63.9** | 59.6 | 52.7 | 86.3 | 69.8 | 1509.1 | **67.4** | 68.5 ↑2.0% |
| HAWAII [NeurIPS'25] | 79.1 | 62.8 | 58.7 | **53.9** | **87.3** | 70.5 | **1540.2** | 66.9 | 68.5 ↑2.0% |
| **GRACE (Ours)** | 79.7 | 63.3 | **61.5** | 52.9 | 86.2 | **71.7** | 1525.8 | 66.6 | **69.0** ↑2.5% |

*Table 2.* Comparison with quantization methods on LLaVA-1.5-7B. Best results among INT4 models are **bolded**, second best are underlined.

| Bitwidth | Method | VQA$^{v2}$ | GQA | TextVQA | VizWiz | POPE | SQA | MMB | Avg. |
|---|---|---|---|---|---|---|---|---|---|
| BF16 | LLaVA-1.5-13B (Teacher) | 80.0 | 63.5 | 61.3 | 53.6 | 85.9 | 71.6 | 67.7 | 69.1 |
| BF16 | LLaVA-1.5-7B (Baseline) | 78.5 | 62.0 | 58.2 | 50.0 | 85.9 | 66.8 | 64.3 | 66.5 |
| 8-bit | **GRACE (Ours)** | 79.2 | 62.8 | 60.4 | 52.5 | 85.9 | 71.3 | 66.1 | 68.3 ↑1.8% |
| 4-bit | RTN | 76.4 | 61.2 | 57.6 | 48.5 | 84.7 | 65.3 | 62.6 | 65.2 ↓1.3% |
| 4-bit | AWQ [MLSys'24] | 77.0 | 61.3 | 57.2 | 49.3 | 85.1 | 66.2 | 63.1 | 65.6 ↓0.9% |
| 4-bit | QAT | 77.3 | **61.5** | 57.3 | 49.8 | 85.0 | 66.5 | 63.9 | 65.9 ↓0.6% |
| 4-bit | **GRACE (Ours)** | **78.3** | 61.2 | **59.2** | **51.6** | **85.2** | **70.1** | **65.0** | **67.2** ↑0.7% |

## 3.5. Adaptive IB Controller

Having established that confidence-gated distillation maximizes $I(Z_S; Y_T)$ (Section 3.2) while quantization constrains $I(Z_S; X)$ (Section 3.4), we now derive a principled mechanism to balance these competing objectives. Since directly estimating mutual information is intractable for high-dimensional vocabularies, we employ $\mathcal{L}_{\text{GDKD}}$ as a surrogate, justified by Proposition 3.2 and consistent with practical IB applications (Alemi et al., 2016; Fischer, 2020). This yields the following constrained optimization formulation:

$$\min_\theta \mathcal{L}_{\text{CE}}(\theta) \quad \text{s.t.} \quad \mathcal{L}_{\text{GDKD}}(\theta) \leq \tau \qquad (18)$$

where $\tau$ serves as the *information preservation budget*, controlling the minimum amount of teacher knowledge to be retained. The Lagrangian relaxation yields:

$$\mathcal{L}(\theta, \beta) = \mathcal{L}_{\text{CE}}(\theta) + \beta \left( \mathcal{L}_{\text{GDKD}}(\theta) - \tau \right), \quad \beta \geq 0 \quad (19)$$

Rather than treating $\beta$ as a fixed hyperparameter, we update it via projected dual ascent with EMA smoothing:

$$\beta \leftarrow \Pi_{[\beta_{\min}, \beta_{\max}]} \left( \beta + \eta \cdot (\widehat{\mathcal{L}}_{\text{GDKD}} - \tau) \right) \qquad (20)$$

This update rule implements intuitive feedback: when $\widehat{\mathcal{L}}_{\text{GDKD}} > \tau$, the student struggles to retain teacher knowledge, prompting $\beta$ to increase; conversely, when the constraint is satisfied, $\beta$ decreases to prioritize task performance.

The complete GRACE training objective combines all components:

$$\mathcal{L}_{\text{total}} = \mathcal{L}_{\text{CE}} + \beta \cdot \mathcal{L}_{\text{GDKD}} + \omega \cdot \mathcal{L}_{\text{RCKA}} \qquad (21)$$

where $\beta$ adapts dynamically via Eq. (20) and $\omega$ is a fixed weight for relational alignment. A detailed derivation connecting this formulation to IB theory is provided in Appendix B.4.

## 4. Experiments

**Models and Configurations.** We evaluate GRACE on two representative VLM families: LLaVA-1.5 (Liu et al., 2024a), using the 13B model as the teacher and the 7B variant as the student, and Qwen2-VL (Wang et al., 2024b), leveraging the 7B model as the teacher and a 2B model as the student. For each family, we conducted experiments in three settings: full-precision distillation, INT8 quantization, and INT4 quantization. In all cases, the models used for distillation are initialized from their pretrained weights and then fine-tuned with our distillation objectives. Following standard practice in VLMs finetuning (Liu et al., 2023), we freeze the vision encoder and maintain it in BF16 precision throughout training. Quantization is applied exclusively to the weights of all linear layers within the LLM backbone, using symmetric group-wise quantization.

**Training and Evaluation Setup.** Training is conducted using the ShareGPT4V dataset (Chen et al., 2024a), which contains 1.3M high-quality image-text pairs generated by GPT-4V and an extended caption model. The models are trained on 8 NVIDIA H100, with 12 hours of training for LLaVA-1.5 and 8 hours for Qwen2-VL. We evaluate models on diverse benchmarks, which include comprehensive multimodal evaluation, visual reasoning, text recognition,

*Table 3.* Ablation study on distillation components. GDKD: Confidence-Gated Decoupled Knowledge Distillation; RCKA: Relational Centered Kernel Alignment; Adaptive IB: Adaptive Information Bottleneck Controller.

| GDKD | RCKA | Adaptive IB | MMBench | SEED-Bench | ScienceQA | Avg. |
|---|---|---|---|---|---|---|
| | Qwen2-VL-7B (Teacher) | | 80.7 | 76.9 | 83.3 | 80.3 |
| | Qwen2-VL-2B (Baseline) | | 71.6 | 72.7 | 73.7 | 72.7 |
| ✗ | ✗ | ✗ | 74.2 | 73.3 | 76.8 | 74.8 (+2.1%) |
| ✓ | ✗ | ✗ | 76.1 | 74.2 | 79.4 | 76.6 (+3.9%) |
| ✗ | ✓ | ✗ | 76.5 | 74.7 | 79.2 | 76.8 (+4.1%) |
| ✗ | ✗ | ✓ | 75.3 | 73.7 | 77.6 | 75.5 (+2.8%) |
| ✓ | ✓ | ✗ | 77.1 | 75.1 | 80.2 | 77.5 (+4.8%) |
| ✓ | ✗ | ✓ | 76.8 | 74.6 | 79.9 | 77.1 (+4.4%) |
| ✗ | ✓ | ✓ | 76.9 | 75.0 | 79.8 | 77.2 (+4.5%) |
| ✓ | ✓ | ✓ | **77.9** | **75.7** | **81.0** | **78.2** (+5.5%) |

*Table 4.* Ablation study on quantization. We compare QAT alone, QAT with naive KD, and QAT with GRACE.

| Precision | KD Method | MMBench | SEED-Bench | ScienceQA | Avg. |
|---|---|---|---|---|---|
| BF16 | Qwen2-VL-7B (Teacher) | 80.7 | 76.9 | 83.3 | 80.3 |
| BF16 | Qwen2-VL-2B (Baseline) | 71.6 | 72.7 | 73.7 | 72.7 |
| 8-bit | QAT only | 71.2 | 72.3 | 73.0 | 72.2 (−0.5%) |
| 8-bit | + Naive KD | 73.8 | 73.2 | 76.3 | 74.4 (+1.7%) |
| 8-bit | + GRACE | **77.4** | **75.5** | **80.8** | **77.9** (+5.2%) |
| 4-bit | QAT only | 70.4 | 71.4 | 72.2 | 71.3 (−1.4%) |
| 4-bit | + Naive KD | 73.0 | 72.1 | 75.2 | 73.4 (+0.7%) |
| 4-bit | + GRACE | **76.9** | **75.6** | **79.1** | **77.2** (+4.5%) |

visual question answering, visual perception, and hallucination evaluation,detailed experimental settings and data compositions are provided in Appendix D. All inference experiments are conducted on a single NVIDIA A100 GPU.

## 4.1. Main Results

We evaluate GRACE on: LLaVA-1.5 and Qwen2-VL. For LLaVA-1.5, we use the 13B model as teacher and 7B as student; for Qwen2-VL, we use 7B as teacher and 2B as student. The main text presents LLaVA-1.5 results; Qwen2-VL experiments are provided in Appendix A.1 (Table 5).

**Knowledge Distillation Performance.** Table 1 compares GRACE with state-of-the-art knowledge distillation methods on LLaVA-1.5. GRACE achieves 69.0% average accuracy, representing a **2.5% improvement over the baseline** that closely approaches the 13B teacher (69.1%). Notably, GRACE outperforms both MoVE-KD-v1.1 (Cao et al., 2025) and HAWAII (Wang et al., 2025) by 0.5% absolute, despite these methods employing more complex architectures. MoVE-KD uses multi-teacher distillation with LoRA mixture-of-experts, while HAWAII incorporates multiple vision encoders. In particular, GRACE achieves substantial gains on TextVQA (+3.2% over MoVE-KD, +2.8% over HAWAII) and ScienceQA (+2.4% over MoVE-KD, +1.2% over HAWAII). These results validate that confidence-gated distillation combined with relational alignment can effectively transfer teacher knowledge without requiring complex multi-teacher frameworks.

**Quantization Performance.** Table 2 evaluates GRACE un-der different bit-widths against PTQ and QAT baselines. At 8-bit precision, GRACE achieves 68.3%, surpassing the full-precision baseline by 1.8%. At 4-bit, PTQ methods exhibit significant degradation (RTN: −1.3%, AWQ: −0.9%), while vanilla QAT only partially recovers performance (65.9%, −0.6%). In contrast, **our 4-bit GRACE attains 67.2%, exceeding the BF16 baseline by 0.7%** and outperforming AWQ and QAT by 1.6% and 1.3% absolute, respectively. These results demonstrate that joint optimization of distillation and quantization enables representations inherently robust to aggressive low-bit compression.

## 4.2. Ablation Studies

We conduct ablation studies to validate each component in GRACE. All experiments use Qwen2-VL (7B→2B) evaluated on MMBench (Liu et al., 2024b), SEED-Bench (Li et al., 2023a), and ScienceQA (Lu et al., 2022).

**Effect of Distillation Components.** Table 3 isolates the contribution of each distillation component. Starting from vanilla KD (74.8%), adding either GDKD or RCKA yields clear gains, improving the average accuracy to 76.6% and 76.8%, respectively. The strong performance of GDKD confirms that teacher confidence is an effective signal for filtering unreliable token-level supervision, while the gain from RCKA shows that preserving visual-token relational structure is crucial for VLM distillation. The adaptive IB controller alone brings a moderate improvement to 75.5%, as its role is mainly to regulate the strength of distillation rather than introduce new supervision. Combining all compo-

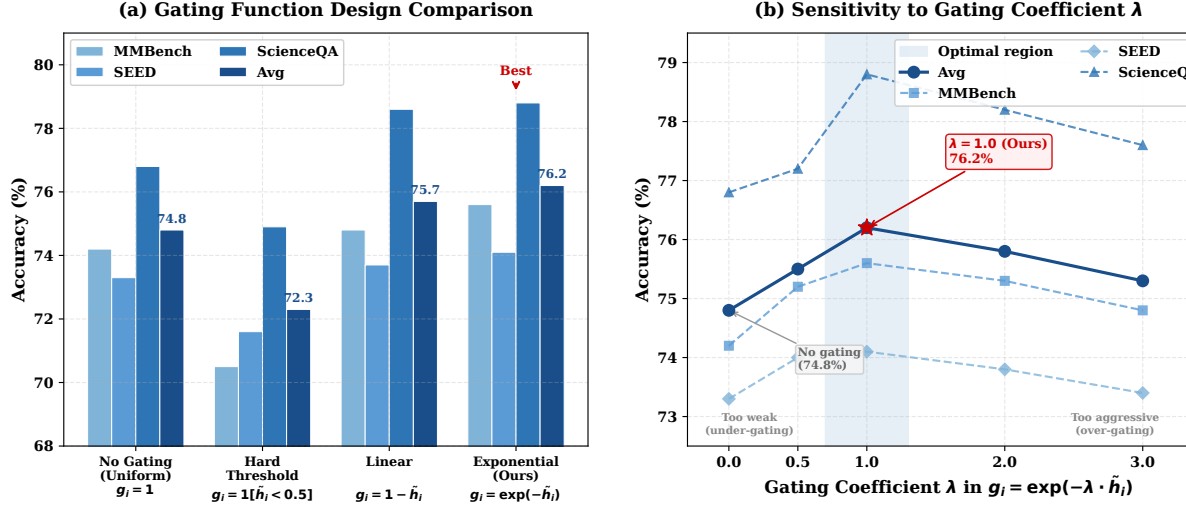

Figure 6. Ablation study on confidence gating design under standard KL distillation on Qwen2-VL (7B→2B, BF16). Here $\tilde{h}_i = H(P_T^{(i)})/\log|V|$ denotes normalized teacher entropy.

nents achieves the best result of 78.2%, demonstrating that confidence-aware logit transfer, relational structure alignment, and adaptive regularization provide complementary benefits.

**Effect of Quantization Components.** Table 4 validates the necessity of jointly optimizing quantization and distillation. QAT alone falls below the BF16 baseline at both 8-bit ($-0.5\%$) and 4-bit ($-1.4\%$), indicating that learned scales and weight adaptation are insufficient to fully recover information lost under low-bit constraints. Adding naive KD partially mitigates this degradation, but its gains remain limited because all teacher predictions are treated equally, including uncertain or noisy ones. In contrast, QAT combined with GRACE achieves substantially larger improvements, reaching 77.9% at 8-bit and 77.2% at 4-bit. The smaller gap between 8-bit and 4-bit under GRACE suggests that confidence-gated and relational supervision help the student preserve task-critical information even under aggressive quantization.

**Effect of Gating Design.** Figure 6 further compares different confidence gating functions under standard KL distillation. Without gating, the student reaches 74.8%, showing that teacher supervision is useful but still contains noisy signals. Hard thresholding drops sharply to 72.3%, even below the no-gating baseline, suggesting that binary filtering removes not only unreliable tokens but also partially informative ones. Linear gating improves over no gating by softly reducing the influence of high-entropy predictions. Exponential gating performs best when $\lambda = 1.0$, reaching 76.2%, because it provides a smoother confidence-dependent weighting while retaining useful supervision from moderately uncertain tokens. The results also show an inverted-U trend: $\lambda = 0.5$ under-suppresses noisy teacher predictions, whereas $\lambda = 3.0$ over-suppresses supervision

and hurts knowledge transfer. This supports our design choice of entropy-based soft gating for robust distillation.

## 5. Conclusion

We presented GRACE, a unified framework for efficient VLM compression that formulates QAT-based low-bit training through an Information Bottleneck perspective. By using the teacher as a proxy for task-relevant information, GRACE guides the quantized student to allocate its limited capacity toward reliable and structurally meaningful knowledge. Through confidence-gated decoupled distillation, relational CKA, adaptive IB control, and group-wise learned step-size quantization, GRACE substantially improves both distillation and quantization performance. Experiments on LLaVA-1.5 and Qwen2-VL show that GRACE enables INT4 models to *surpass* their BF16 baselines, demonstrating the effectiveness of principled information allocation over standard QAT and naive QAT-KD combinations.

Despite these promising results, GRACE still has several limitations. First, our current implementation focuses on weight-only quantization, while activation quantization remains unexplored and may introduce additional instability in multimodal reasoning. Second, the teacher-student setting requires access to a stronger teacher model during training, which increases offline training cost. Third, our experiments mainly cover image-text VLMs, leaving broader multimodal architectures, such as video-language and audio-language models, for future investigation. Future work will extend GRACE to weight-activation quantization, reduce the dependency on large teachers, and explore its applicability to more diverse multimodal systems.

## Acknowledgement

This work was supported by a grant from the Swiss National Supercomputing Centre (CSCS) under project IDs lp12 and lp160 on Alps. We acknowledge the CINECA award under the ISCRA initiative (project IsB32) for the availability of HPC resources.

## Impact Statement

This paper presents GRACE, a framework for efficient Vision-Language Model compression through joint knowledge distillation and quantization-aware training. Our work aims to advance the field of Machine Learning by enabling the deployment of high-quality VLMs on resource-constrained devices, thereby democratizing access to multimodal AI capabilities.

We acknowledge several potential societal implications of our work. On the positive side, efficient VLM compression can reduce computational resources and energy consumption required for inference, contributing to a more sustainable AI deployment. Making powerful vision-language models accessible on edge devices can benefit applications in education, accessibility tools, and healthcare in resource-limited settings.

However, as with any advancement in AI capabilities, there is a potential for misuse. More accessible VLMs could be employed to generate misleading content or for surveillance applications. We encourage the research community and practitioners to deploy compressed VLMs responsibly, with appropriate safeguards, and in compliance with ethical guidelines.

In general, we believe that the benefits of democratizing access to efficient multimodal AI outweigh the risks, provided that deployment is conducted with appropriate consideration for ethical implications.

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

# A. Additional Experiments

*Table 5.* Comparison with quantization methods on Qwen2-VL-2B. Best results among 4-bit models are **bolded**, second best are underlined.

| Bitwidth | Method | MMB | MMStar | MMMU | Hallusion | AI2D | OCR | SEED | SQA | Avg. |
|---|---|---|---|---|---|---|---|---|---|---|
| BF16 | Qwen2-VL-7B (Teacher) | 80.7 | 60.7 | 54.1 | 50.6 | 83.0 | 84.5 | 76.9 | 83.3 | 71.7 |
| BF16 | Qwen2-VL-2B (Baseline) | 72.6 | 50.4 | 45.4 | 42.9 | 73.1 | 81.0 | 72.7 | 73.7 | 64.0 |
| BF16 | **GRACE (Ours)** | 77.9 | 55.5 | 52.0 | 48.3 | 80.0 | 83.1 | 75.7 | 81.0 | **69.2** ↑5.2% |
| 8-bit | **GRACE (Ours)** | 77.4 | 55.2 | 51.8 | 47.9 | 79.7 | 82.9 | 75.5 | 80.8 | **68.9** ↑4.9% |
| 4-bit | RTN | 70.98 | 45.07 | 37.22 | 42.74 | 71.15 | 78.8 | 72.45 | 70.67 | 61.1 ↓2.9% |
| 4-bit | GPTQ [ICLR'23] | 70.51 | 46.33 | 36.22 | 39.62 | 70.53 | 79.5 | 72.62 | 72.10 | 60.9 ↓3.1% |
| 4-bit | AWQ [MLSys'24] | 68.89 | 44.80 | 37.33 | 39.55 | 70.08 | 78.9 | 71.83 | 72.15 | 60.4 ↓3.6% |
| 4-bit | MBQ [CVPR'25] | 70.55 | 44.53 | 38.22 | 39.89 | 70.21 | 80.9 | 71.86 | 71.72 | 61.0 ↓3.0% |
| 4-bit | SPEED-Q [arXiv'25] | 69.85 | 50.87 | 42.0 | 41.71 | 70.92 | 76.5 | 74.40 | 72.01 | 62.3 ↓1.7% |
| 4-bit | **GRACE (Ours)** | **76.9** | **54.3** | **51.1** | **46.5** | **78.6** | **81.4** | **75.6** | **79.1** | **68.0** ↑4.0% |

## A.1. Generalization to the Qwen2-VL

Table 5 demonstrates that GRACE generalizes beyond LLaVA to different VLM architectures. On Qwen2-VL (7B→2B), full-precision distillation improves the student by **5.2%**, showing strong capacity to bridge larger teacher-student gaps. Notably, the distilled student recovers 96.5% of the teacher's performance (69.2 vs. 71.7), demonstrating effective knowledge transfer even with a 3.5× parameter reduction.

For 4-bit quantization, PTQ methods (RTN, GPTQ, AWQ, MBQ) suffer 2.9–3.6% degradation, with particularly severe drops on reasoning-intensive benchmarks: MMMU drops from 45.4 to as low as 36.2 under GPTQ. SPEED-Q (Guo et al., 2025), a recent QAT method specifically designed to address vision-language quantization sensitivity, still incurs a 1.7% performance drop. In contrast, **GRACE achieves 68.0% (+4.0% over baseline)**, outperforming SPEED-Q by **5.7 percentage points**. The improvement is consistent across all benchmarks, with the largest gains on MMMU (+9.1 over baseline) and MMStar (+3.4), both of which require complex multimodal reasoning. Furthermore, our 4-bit model nearly matches the 8-bit performance (68.0 vs. 68.9), indicating that GRACE effectively preserves critical information even under aggressive quantization. This highlights our advantage: rather than treating quantization as a module-specific problem, GRACE leverages teacher guidance to learn globally coherent, quantization-robust representations.

*Table 6.* **PTQ on a GRACE-distilled BF16 model vs. our joint QAT+KD pipeline** on Qwen2-VL (7B→2B). Naive PTQ erodes the distillation gain, whereas joint optimization preserves nearly all of it at INT4. Best per column in **bold**; our method shaded.

| Pipeline | Precision | MMMU | DocVQA | MathVista | Avg. |
|---|---|---|---|---|---|
| *Standard pipeline (no distillation)* | | | | | |
| Baseline | BF16 | 45.4 | 88.3 | 44.4 | 59.4 |
| ↪ GPTQ | INT4 | 36.2 | 87.2 | 41.7 | 55.0 |
| ↪ AWQ | INT4 | 37.3 | 87.0 | 39.9 | 54.7 |
| *GRACE pipeline (with knowledge distillation)* | | | | | |
| GRACE (KD only) | BF16 | **52.0** | **90.5** | **50.2** | **64.2** |
| ↪ GPTQ | INT4 | 46.5 | 89.2 | 46.9 | 60.9 |
| ↪ AWQ | INT4 | 47.2 | 89.1 | 45.1 | 60.5 |
| **GRACE (Joint QAT+KD, ours)** | INT4 | 51.1 | 90.2 | 49.5 | 63.6 |

## A.2. Post-training Quantization after GRACE Distillation

We further investigate whether the robustness gained from GRACE distillation can be preserved by standard post-training quantization. Specifically, we apply GPTQ-Int4 and AWQ-Int4 to the GRACE-distilled BF16 model and compare them with

our joint QAT+KD pipeline. All experiments are conducted on Qwen2-VL (7B→2B).

Table 6 shows that GRACE distillation improves PTQ robustness, but does not replace joint optimization. Applying GPTQ/AWQ to the BF16 baseline gives an average accuracy of about 54.9%, while applying the same PTQ methods after GRACE BF16 improves the average to about 60.7%, a gain of 5.8%. This confirms that GRACE distillation makes the model more resilient to subsequent quantization. However, our joint GRACE INT4 pipeline still achieves 63.6%, outperforming the best GRACE→PTQ result by 2.7%. Moreover, joint QAT+KD loses only 0.6% from GRACE BF16, whereas the PTQ pipelines lose 3.3–3.7%. These results indicate that distillation can improve post-hoc quantizability, but explicitly training under the quantization constraint is still necessary to allocate the limited bit budget toward task-relevant information.

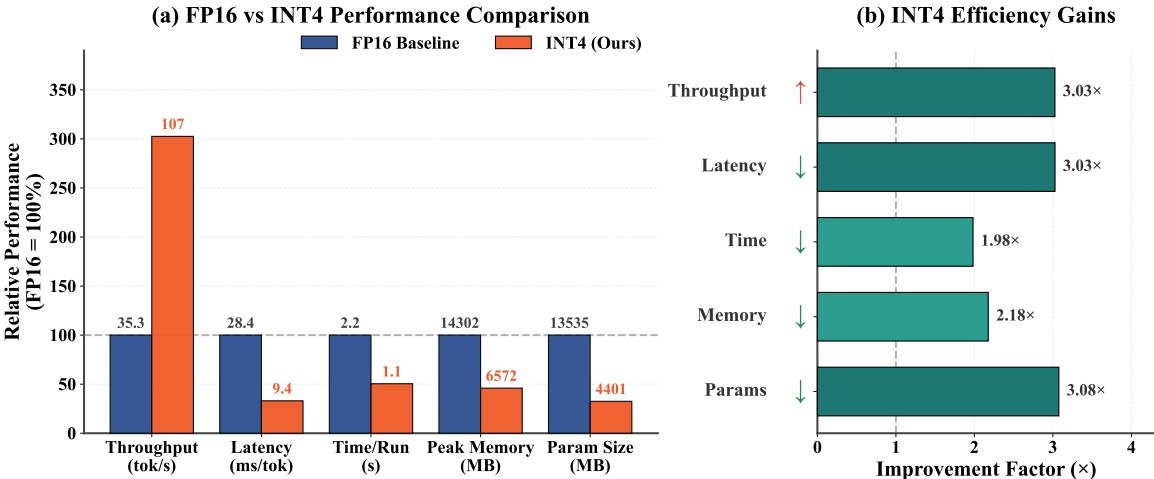

*Figure 7.* Deployment efficiency comparison between LLaVA-1.5 7B FP16 and GRACE 7B INT4 on A100 GPU. (a) Performance metrics normalized to FP16 baseline (100%). (b) Improvement factors achieved by INT4 quantization.

### A.3. Deployment Efficiency

To evaluate practical deployment benefits, we benchmark GRACE 7B INT4 against LLaVA-1.5 7B FP16 on a single NVIDIA A100 GPU. For INT4 inference, we employ TinyChat (Lin et al., 2024), which provides optimized INT4 CUDA kernels including fused attention, fused MLP, and fused normalization layers. These kernels minimize memory bandwidth overhead and maximize arithmetic intensity. Unlike simulated quantization, which performs computation in higher precision, our deployment uses actual INT4 matrix multiplication with group-wise quantization (group size 128), reflecting true inference performance.

We measure throughput (tokens/s), latency (ms/token), peak GPU memory, and model parameter size. Each metric is averaged over multiple runs with warmup iterations to ensure statistical stability.

As shown in Figure 7, GRACE 7B INT4 delivers substantial efficiency gains. For inference speed, our INT4 model achieves **3.03× higher throughput** (106.68 vs 35.26 tokens/s) and **3.03× lower latency** (9.37 vs 28.36 ms/token). This speedup stems from reduced memory bandwidth requirements and the efficiency of TinyChat's fused CUDA kernels. For memory efficiency, GRACE reduces **peak GPU memory by 2.18×** (6,572 MB vs 14,302 MB) and **parameter storage by 3.08×** (4,401 MB vs 13,535 MB). These reductions enable deployment on resource-constrained devices or support larger batch sizes, demonstrating that GRACE delivers significant practical benefits alongside preserved accuracy (Section 4.1).

## B. Theoretical Proofs

This appendix provides detailed proofs for the theoretical results presented in Section 3.

### B.1. Fano's Inequality and Confidence-Gated Distillation

In Section 2.1, we empirically observed a strong correlation between teacher entropy and prediction error rate. Here we provide the theoretical foundation for this observation through Fano's inequality.

**Theorem B.1** (Fano's Inequality). *Let $X$ be a discrete random variable taking values in $\mathcal{X}$ with $|\mathcal{X}| = K$, and let $\hat{X}$ be an estimate of $X$ based on an observation $Y$. Define the error probability $P_e = \Pr(\hat{X} \neq X)$. Then:*

$$H(X|Y) \leq H(P_e) + P_e \log(K - 1) \tag{22}$$

*where $H(P_e) = -P_e \log P_e - (1 - P_e) \log(1 - P_e)$ is the binary entropy function.*

Rearranging this inequality yields a lower bound on the error probability:

$$P_e \geq \frac{H(X|Y) - 1}{\log K} \tag{23}$$

**Connection to Knowledge Distillation.** In the context of knowledge distillation, we can interpret the teacher's output distribution $P_T$ as providing a noisy observation of the true label $X$. The entropy of the teacher's prediction, $H(P_T) = -\sum_v P_T(v) \log P_T(v)$, serves as a proxy for the conditional entropy $H(X|Y)$. According to Eq. 23, higher entropy in the teacher's output necessarily implies a higher lower bound on the probability of prediction error.

This theoretical result provides information-theoretic justification for our confidence-gated distillation mechanism: when the teacher exhibits high entropy (uncertainty), Fano's inequality guarantees that the error probability is bounded away from zero. Consequently, such predictions are inherently unreliable supervision signals, and down-weighting them during distillation prevents the student from learning from noisy or erroneous teacher outputs.

**Empirical Validation.** Our experiments in Section 2.1 confirm this theoretical prediction. The strong correlation between teacher entropy and error rate (Pearson $r = 0.484$, binned $R^2 = 0.901$) demonstrates that the bound established by Fano's inequality manifests as a practical relationship in VLM distillation, validating the design of our confidence gating function $g(\cdot)$ in Eq. 5.

### B.2. Proof of Theorem 3.1: Effect of Confidence Gating

**Theorem** (Restated). *Let $g_i = \exp(-\tilde{h}_i)$ and $w_i = g_i / \sum_{j=1}^{N} g_j$ be the normalized confidence weights satisfying $\sum_{i=1}^{N} w_i = 1$. Let $L_i \triangleq \mathcal{L}_{DKD}^{(i)}$ denote the per-token DKD loss and $\bar{L} \triangleq \frac{1}{N} \sum_{i=1}^{N} L_i$ its unweighted average. Define the sample covariance under the uniform distribution over token indices:*

$$\mathrm{Cov}(w_i, L_i) \triangleq \frac{1}{N} \sum_{i=1}^{N} (w_i - \bar{w})(L_i - \bar{L}), \qquad \bar{w} \triangleq \frac{1}{N} \sum_{i=1}^{N} w_i = \frac{1}{N}. \tag{24}$$

*Then the gated loss satisfies the exact identity*

$$\mathcal{L}_{GDKD} = \bar{L} + N \cdot \mathrm{Cov}(w_i, L_i). \tag{25}$$

*Moreover, under the monotonicity assumption in Eq. (30), we have $\mathrm{Cov}(w_i, L_i) \leq 0$ and thus $\mathcal{L}_{GDKD} \leq \bar{L}$.*

*Proof.* We proceed in four steps.

**Step 1: Setup.** By definition of confidence gating,

$$\mathcal{L}_{\mathrm{GDKD}} = \frac{\sum_{i=1}^{N} g_i L_i}{\sum_{i=1}^{N} g_i} = \sum_{i=1}^{N} w_i L_i, \qquad w_i = \frac{g_i}{\sum_{j=1}^{N} g_j}, \qquad \sum_{i=1}^{N} w_i = 1. \tag{26}$$

**Step 2: Exact covariance decomposition.** Using the definition of sample covariance and $\bar{w} = \frac{1}{N}$,

$$\mathrm{Cov}(w_i, L_i) = \frac{1}{N} \sum_{i=1}^{N} (w_i - \bar{w})(L_i - \bar{L})$$
$$= \frac{1}{N} \sum_{i=1}^{N} w_i L_i - \bar{w}\bar{L}, \tag{27}$$

where the cross terms vanish since $\sum_i(w_i - \bar{w}) = 0$ and $\sum_i(L_i - \bar{L}) = 0$. Since $\bar{w} = \frac{1}{N}$, multiplying Eq. (27) by $N$ yields

$$N \cdot \text{Cov}(w_i, L_i) = \sum_{i=1}^{N} w_i L_i - \bar{L}. \tag{28}$$

Combining Eq. (28) with Eq. (26) gives the desired identity:

$$\boxed{\mathcal{L}_{\text{GDKD}} = \sum_{i=1}^{N} w_i L_i = \bar{L} + N \cdot \text{Cov}(w_i, L_i)} \tag{29}$$

which proves Eq. (25).

**Step 3: Sufficient condition for $\text{Cov}(w_i, L_i) \leq 0$.** We impose the following monotonicity assumption:

**Assumption B.2** (Entropy–Loss Monotonicity). The per-token DKD loss is weakly non-decreasing with respect to teacher normalized entropy:

$$(\tilde{h}_i - \tilde{h}_j)(L_i - L_j) \geq 0, \qquad \forall\, i, j \in \{1, \dots, N\}. \tag{30}$$

Since $g_i = \exp(-\tilde{h}_i)$ is strictly decreasing in $\tilde{h}_i$ and the normalization constant $\sum_j g_j$ is positive and independent of index ordering, we have the opposite ordering between $w_i$ and $\tilde{h}_i$:

$$\tilde{h}_i \geq \tilde{h}_j \implies g_i \leq g_j \implies w_i \leq w_j. \tag{31}$$

To determine the sign of the covariance, we use the pairwise form:

$$\text{Cov}(w_i, L_i) = \frac{1}{2N^2} \sum_{i=1}^{N} \sum_{j=1}^{N} (w_i - w_j)(L_i - L_j). \tag{32}$$

This identity can be verified by expanding the double sum: $\sum_{i,j}(w_i - w_j)(L_i - L_j) = 2N \sum_i w_i L_i - 2(\sum_i w_i)(\sum_i L_i)$. Now consider any pair $(i, j)$:

- If $\tilde{h}_i \geq \tilde{h}_j$, then by Eq. (30), $L_i \geq L_j$, and by Eq. (31), $w_i \leq w_j$. Thus $(w_i - w_j)(L_i - L_j) \leq 0$.

- If $\tilde{h}_i \leq \tilde{h}_j$, the inequalities reverse, and we still have $(w_i - w_j)(L_i - L_j) \leq 0$.

Hence every term in Eq. (32) is non-positive, implying

$$\boxed{\text{Cov}(w_i, L_i) \leq 0} \tag{33}$$

Furthermore, if there exists at least one pair $(i, j)$ such that $\tilde{h}_i \neq \tilde{h}_j$ and $L_i \neq L_j$, then at least one term is strictly negative, and thus $\text{Cov}(w_i, L_i) < 0$.

**Step 4: Conclusion.** Substituting $\text{Cov}(w_i, L_i) \leq 0$ into Eq. (25) yields

$$\boxed{\mathcal{L}_{\text{GDKD}} = \bar{L} + N \cdot \text{Cov}(w_i, L_i) \leq \bar{L}} \tag{34}$$

with strict inequality whenever $\text{Cov}(w_i, L_i) < 0$. This completes the proof. $\qquad \square$

**Remark.** When $\tilde{h}_i$ and $\mathcal{L}_{\text{DKD}}^{(i)}$ are positively correlated (validated empirically in Figure 2), high-entropy tokens tend to have high loss. Since $w_i$ assigns lower weights to these tokens, the covariance $\text{Cov}(w_i, \mathcal{L}_{\text{DKD}}^{(i)})$ is negative, and thus $\mathcal{L}_{\text{GDKD}} < \bar{\mathcal{L}}_{\text{DKD}}$. This confirms that gating reduces effective distillation pressure on unreliable predictions and concentrates gradients on confident supervision.

**Interpretation.**  The theorem provides an exact characterization of confidence gating without requiring any approximation. The gated loss differs from the uniform average by a term proportional to the covariance between normalized weights and per-token losses. When high-entropy tokens (which receive lower weights) also exhibit higher distillation loss, this covariance is negative, causing the gated loss to be smaller than the unweighted average. This confirms that entropy-based gating automatically identifies and down-weights supervision signals that are unreliable (high entropy) and difficult to match (high loss), thereby improving distillation quality.

### B.3. Proof of Proposition 3.2: Variational Lower Bound and KL Gap

**Assumptions and Justification.**  Our analysis relies on two structural assumptions that we state explicitly:

**Assumption B.3** (Deterministic Encoder)**.**  The student representation $Z_S = f_S(X)$ is a deterministic function of the input.

**Assumption B.4** (Sufficient Statistic Decoder)**.**  The student prediction depends on input only through its representation: $P_S(y \mid X) = P_S(y \mid Z_S)$.

**Justification.** These assumptions hold *by construction* in standard VLM architectures. The encoder $f_S$ (comprising the vision encoder, projection layer, and LLM backbone) deterministically maps inputs to hidden states. The language model head then produces output distributions solely from these final-layer representations, satisfying Assumption B.4. Together, they imply the Markov chain $Y_T \to X \to Z_S$, which is the data processing inequality's precondition. This structure is not a restrictive modeling choice but rather an accurate description of how modern transformer-based VLMs operate.

**Proposition** (Restated)**.**  *Let $Y_T$ be a pseudo-label sampled from the teacher, i.e., $Y_T \mid X = x \sim P_T(\cdot \mid x)$. Let $Z_S = f_S(X)$ denote the student's representation. Under Assumptions B.3 and B.4, we have*

$$I(Z_S; Y_T) \geq I(X; Y_T) - \mathbb{E}_X[D_{\mathrm{KL}}(P_T(\cdot \mid X) \,\|\, P_S(\cdot \mid X))]. \tag{35}$$

*Proof.*  We proceed in six steps.

**Step 1: Mutual information definition.**  By definition,

$$I(Z_S; Y_T) = H(Y_T) - H(Y_T \mid Z_S). \tag{36}$$

**Step 2: Variational upper bound on $H(Y_T \mid Z_S)$.**  For any conditional distribution $q(y \mid z)$,

$$\begin{aligned}
H(Y_T \mid Z_S) &= \mathbb{E}_{Y_T, Z_S}\big[ -\log p(Y_T \mid Z_S)\big] \\
&\leq \mathbb{E}_{Y_T, Z_S}\big[ -\log q(Y_T \mid Z_S)\big],
\end{aligned} \tag{37}$$

since $\mathbb{E}_{Z_S}[D_{\mathrm{KL}}(p(\cdot \mid Z_S) \,\|\, q(\cdot \mid Z_S))] \geq 0$.

**Step 3: Student decoder as variational approximation.**  We choose $q(\cdot \mid Z_S) = P_S(\cdot \mid Z_S)$. Using $Z_S = f_S(X)$ (Assumption B.3) and $Y_T \mid X \sim P_T(\cdot \mid X)$,

$$\begin{aligned}
\mathbb{E}_{Y_T, Z_S}\big[ -\log P_S(Y_T \mid Z_S)\big] &= \mathbb{E}_X \mathbb{E}_{Y_T \mid X}\big[ -\log P_S(Y_T \mid f_S(X))\big] \\
&= \mathbb{E}_X\left[H(P_T(\cdot \mid X),\, P_S(\cdot \mid f_S(X)))\right].
\end{aligned} \tag{38}$$

By Assumption B.4, $P_S(\cdot \mid X) = P_S(\cdot \mid f_S(X))$, so the right-hand side equals $\mathbb{E}_X[H(P_T(\cdot \mid X), P_S(\cdot \mid X))]$. Combining with Eq. (37):

$$\boxed{H(Y_T \mid Z_S) \leq \mathbb{E}_X\left[H(P_T(\cdot \mid X),\, P_S(\cdot \mid X))\right]} \tag{39}$$

**Step 4: Cross-entropy decomposition.**  For each $x$,

$$H(P_T(\cdot \mid x),\, P_S(\cdot \mid x)) = H(P_T(\cdot \mid x)) + D_{\mathrm{KL}}(P_T(\cdot \mid x) \,\|\, P_S(\cdot \mid x)). \tag{40}$$

Taking expectation over $X$:

$$\mathbb{E}_X[H(P_T, P_S)] = \mathbb{E}_X[H(P_T)] + \mathbb{E}_X[D_{\mathrm{KL}}(P_T \| P_S)]. \tag{41}$$

**Step 5: Identifying $H(Y_T \mid X)$.**   Since $Y_T \mid X \sim P_T(\cdot \mid X)$,

$$\mathbb{E}_X[H(P_T(\cdot \mid X))] = H(Y_T \mid X). \tag{42}$$

Substituting into Eq. (39) via Eq. (41):

$$\boxed{H(Y_T \mid Z_S) \le H(Y_T \mid X) + \mathbb{E}_X[D_{\mathrm{KL}}(P_T(\cdot \mid X) \,\|\, P_S(\cdot \mid X))]} \tag{43}$$

**Step 6: Final bound.**   Substituting Eq. (43) into Eq. (36):

$$\begin{aligned}
I(Z_S; Y_T) &\ge H(Y_T) - H(Y_T \mid X) - \mathbb{E}_X[D_{\mathrm{KL}}(P_T \| P_S)] \\
&= I(X; Y_T) - \mathbb{E}_X[D_{\mathrm{KL}}(P_T \| P_S)].
\end{aligned} \tag{44}$$

Therefore,

$$\boxed{I(Z_S; Y_T) \ge I(X; Y_T) - \mathbb{E}_X[D_{\mathrm{KL}}(P_T \| P_S)]} \tag{45}$$

which completes the proof.   □

*Remark* B.5 (Tightness of the Bound). The gap between $I(Z_S; Y_T)$ and its lower bound equals $\mathbb{E}_{Z_S}[D_{\mathrm{KL}}(p(Y_T|Z_S) \| P_S(Y_T|Z_S))]$, the expected divergence between the true posterior $p(Y_T|Z_S)$ and the student's approximation $P_S(Y_T|Z_S)$. In well-trained models where the student closely approximates this posterior, the gap diminishes, making the bound practically tight. Empirically, we observe that converged GRACE models achieve near-zero KL divergence on high-confidence tokens, indicating the bound is approached in practice.

**Interpretation and Connection to GRACE.**   This proposition establishes a principled foundation for KL-based distillation:

1. **Information-theoretic ceiling.** The term $I(X; Y_T)$ represents the maximum mutual information any student could achieve—the information the teacher captures about its own predictions.

2. **KL as information gap.** The expected KL divergence $\mathbb{E}[D_{\mathrm{KL}}(P_T \| P_S)]$ quantifies exactly how much information the student fails to preserve. Perfect matching ($P_S = P_T$) recovers the maximum.

3. **Confidence gating rationale.** Low-entropy (high-confidence) teacher predictions carry more recoverable information per token. By down-weighting high-entropy tokens, confidence gating allocates the quantized student's limited capacity to preserving the most informative supervision signals.

4. **Adaptive $\beta$ as constraint enforcement.** The IB controller directly operationalizes this bound: treating $\mathbb{E}[D_{\mathrm{KL}}]$ as a constraint ensures the student maintains sufficient information about $Y_T$ while the quantization bottleneck limits representation complexity.

## B.4. Theoretical Foundation of the Adaptive IB Controller

This appendix provides a rigorous derivation of the adaptive controller and establishes its connection to the Information Bottleneck (IB) principle.

### B.4.1. FROM INFORMATION BOTTLENECK TO CONSTRAINED OPTIMIZATION

**Mapping IB to GRACE.**   The Information Bottleneck principle (Tishby et al., 2000) seeks representations that maximize task-relevant information while limiting complexity. In GRACE, each IB term admits a concrete realization that directly corresponds to our framework components:

• **Complexity constraint $I(Z_S; X) \le C$:** Quantization *physically* enforces this bound through architectural constraints. Reducing bit-width from 16-bit to $b$-bit limits each layer's information capacity to at most $n \cdot b$ bits, where $n$ is the number of parameters. Unlike soft regularization in standard IB formulations, this constitutes a *hard* constraint that cannot be violated during inference—the discrete nature of quantized weights fundamentally limits the representational capacity.

- **Information preservation** $I(Z_S; Y_T)$: From Proposition 3.2, minimizing $\mathbb{E}[D_{\text{KL}}(P_T \| P_S)]$ maximizes a lower bound on $I(Z_S; Y_T)$. The confidence-gated distillation loss $\mathcal{L}_{\text{GDKD}}$ serves as a tractable surrogate for this KL divergence, with the gating mechanism focusing optimization on tokens where information transfer is most reliable.

This mapping reveals the complementary roles of GRACE's components: quantization handles the compression side of the IB trade-off through hard constraints, while distillation handles the information preservation side through loss minimization.

**Placement of $\beta$ on the Distillation Term.** In the standard IB Lagrangian formulation $\max I(Z; Y) - \beta I(Z; X)$, the multiplier $\beta$ penalizes the complexity term. In GRACE, $\beta$ instead weights the distillation term in Eq. 19. This placement follows naturally from the constrained formulation in Eq. 15: since quantization already enforces $I(Z_S; X) \leq C_b$ as an architectural constraint, explicit penalization of complexity becomes unnecessary.

The optimization focus shifts to ensuring sufficient information preservation under fixed capacity. Directly maximizing $I(Z_S; Y_T)$ without bound, however, can lead to overfitting to teacher outputs at the expense of task performance. The reformulation in Eq. 18 addresses this by minimizing task loss subject to a knowledge retention constraint. The resulting Lagrangian in Eq. 19 places $\beta$ on the distillation term as the dual variable enforcing this constraint. This is not an inversion of standard IB but rather a *complementary* formulation appropriate when the complexity constraint is architecturally enforced.

### B.4.2. PRINCIPLED ADAPTATION VIA DUAL ASCENT

**Theoretical Grounding.** The projected dual ascent update in Eq. 20 implements a principled optimization strategy with well-understood convergence properties. Under mild assumptions (Lipschitz gradients, bounded iterates), projected dual ascent converges to a stationary point of the Lagrangian (Bertsimas & Popescu, 2005). The EMA smoothing ($\widehat{\mathcal{L}}_{\text{GDKD}}$) reduces oscillations from stochastic gradients while preserving these convergence guarantees.

**Advantages over Fixed Weighting.** The constrained optimization perspective provides concrete benefits over heuristic loss weighting schemes:

1. **Interpretable target:** The threshold $\tau$ directly specifies the information preservation budget—the maximum allowable gap between student and teacher knowledge. This interpretability contrasts with arbitrary fixed weights whose optimal values vary across model architectures and datasets.

2. **Automatic adaptation:** The dual ascent mechanism automatically adjusts $\beta$ throughout training. During early stages when the student is far from the teacher, $\widehat{\mathcal{L}}_{\text{GDKD}} > \tau$ triggers increased $\beta$, strengthening distillation pressure. As training progresses and knowledge transfer succeeds, $\beta$ decreases to allow focus on task-specific learning.

3. **Robustness:** Rather than requiring separate hyperparameter searches for $\beta$ across different quantization settings (W4A16, W3A16, etc.), the controller adapts to each setting's capacity constraints. Lower bit-widths naturally require stronger distillation to compensate for reduced capacity, and the controller discovers this automatically.

**Dynamic Behavior During Training.** The interplay between quantization constraints and distillation pressure creates distinctive training dynamics. In early training, quantization-induced capacity limitations cause significant teacher-student divergence, driving $\widehat{\mathcal{L}}_{\text{GDKD}}$ above $\tau$ and increasing $\beta$. As the model adapts its weights to the quantization constraints and learns to preserve critical teacher knowledge within the limited bit-budget, $\widehat{\mathcal{L}}_{\text{GDKD}}$ decreases. Once the constraint is satisfied, reduced $\beta$ allows the cross-entropy loss to guide fine-grained task adaptation. This automatic curriculum—strong distillation early, task focus later—emerges naturally from the optimization dynamics without explicit scheduling.

### B.4.3. UNIFIED IB PERSPECTIVE

The IB framework provides a coherent lens for understanding how GRACE's components address different aspects of efficient VLM compression:

- **Confidence gating** implements selective information transfer. High-entropy teacher predictions contribute less reliable information about the target distribution; down-weighting these tokens focuses the student's limited capacity on preserving the most informative supervision signals, directly supporting the $\max I(Z_S; Y_T)$ objective.

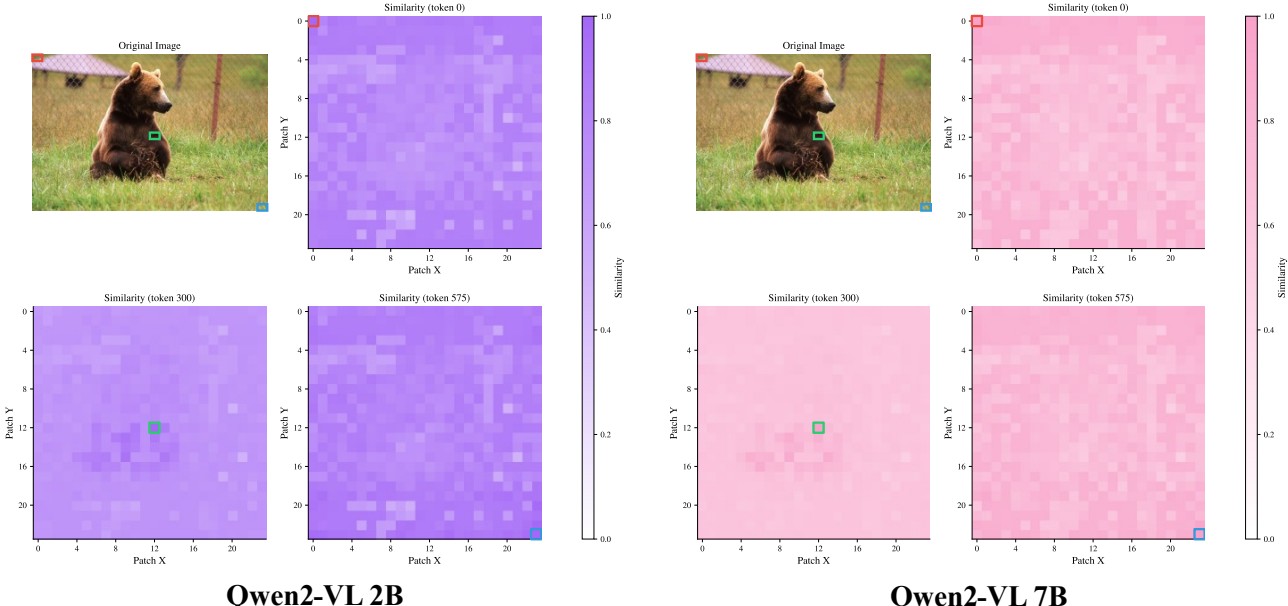

*Figure 8.* Visual token similarity comparison on a bear image. The 7B model shows clearer semantic boundaries, with the center token (green box) activating precisely on the bear region, while the 2B model produces diffuse similarity patterns.

- **Quantization** enforces the complexity constraint $I(Z_S; X) \leq C_b$ through architectural means. The discrete, low-precision weights fundamentally limit how much input information can be encoded, providing the "bottleneck" in the information-theoretic sense.

- **Adaptive** $\beta$ balances the competing objectives dynamically. By treating knowledge retention as a constraint rather than a fixed penalty, the controller ensures sufficient teacher information is preserved ($\mathcal{L}_{\text{GDKD}} \leq \tau$) while maximizing task performance within the quantization-imposed capacity limits.

- **RCKA** complements logit-level distillation by transferring relational structure among visual tokens—information that supports the student's visual reasoning capabilities but is not captured by output distribution matching alone.

This unified perspective demonstrates that GRACE instantiates the IB trade-off through concrete, implementable mechanisms: quantization for compression, confidence-gated distillation for information preservation, and adaptive weighting for principled balancing. The framework transforms abstract information-theoretic objectives into a practical training procedure for efficient VLM compression.

## C. Qualitative Analysis

### C.1. Visual Token Similarity Analysis

We extend the RCKA analysis in Section 3.3 by comparing the visual token similarity patterns between Qwen2-VL 2B and Qwen2-VL 7B models. As shown in Figures 8 and 9, we visualize the pairwise cosine similarity between selected query tokens and all other visual tokens in the spatial grid. For each image, we select three representative tokens: token 0 (top-left corner, typically background), token 300 (center region, typically the main subject), and token 575 (bottom-right corner).

Figure 8 presents the similarity maps for a bear image. In the 2B model, the similarity patterns are diffuse and noisy across all query tokens. When querying the center token (located on the bear), the 2B model produces scattered activations that fail to cleanly segment the bear from the background. In contrast, the 7B model exhibits significantly sharper semantic boundaries: the center token shows high similarity concentrated precisely on the bear region, while tokens in the grass and fence areas form distinct, coherent clusters. This demonstrates that larger models develop more precise relational structures that group semantically related visual tokens.

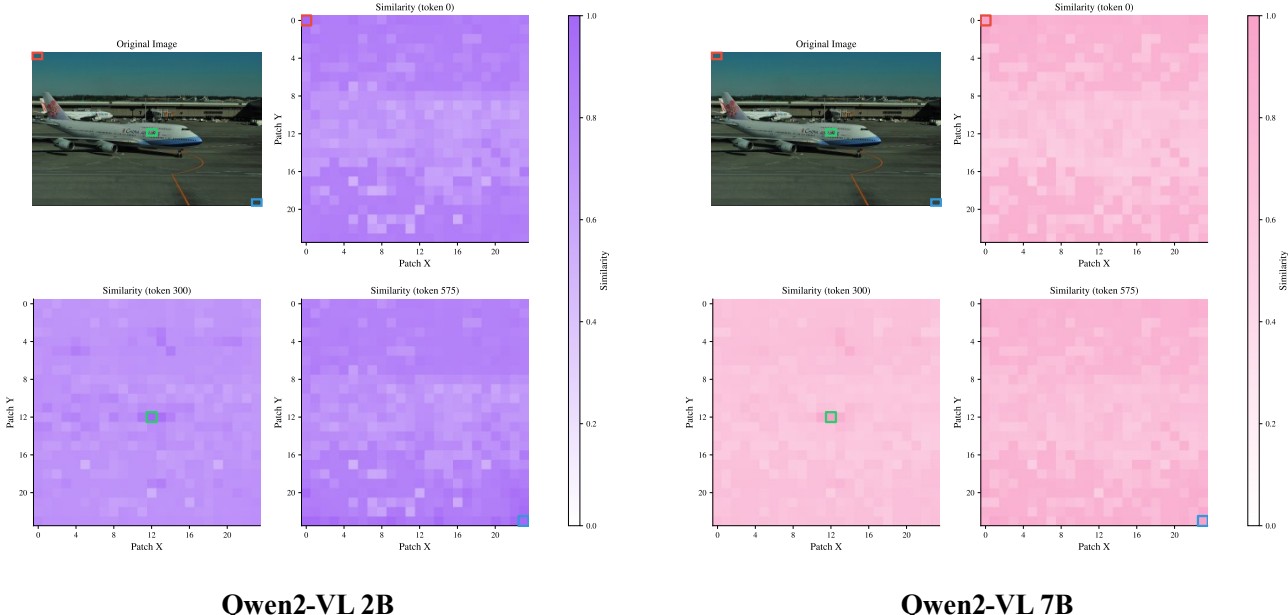

*Figure 9.* Visual token similarity comparison on an airplane image. The 7B model clearly separates sky, airplane, and ground regions, while the 2B model shows diffuse patterns that fail to capture semantic boundaries.

Figure 9 shows a similar pattern for a China Airlines aircraft image. The 2B model struggles to distinguish between the airplane, sky, and ground regions, producing mixed similarity values across semantic boundaries. The 7B model, however, demonstrates clear semantic separation: sky tokens cluster together with high mutual similarity, the airplane fuselage forms a distinct group, and the ground/tarmac region is cleanly segmented. Notably, when querying the center token on the airplane, the 7B model's activation map closely follows the aircraft's shape, indicating that larger models encode fine-grained object boundaries in their relational structure.

These visualizations provide direct motivation for our RCKA loss: by explicitly aligning the student's relational structure with that of the teacher, we transfer the capacity for precise semantic grouping that smaller models inherently lack. The sharper similarity patterns in larger models reflect their superior ability to organize visual information into coherent semantic regions, which is essential for accurate visual reasoning and understanding.

### C.2. Visual Attention Heatmap Analysis

To further understand the differences in visual processing between models of different scales, we visualize the attention patterns from the penultimate layer of the LLM backbone. Specifically, we extract the attention weights from the last generated token attending to all visual tokens, and reshape them into a 2D spatial grid that aligns with the original image. This visualization directly corresponds to the layer we use for RCKA computation (Section 3.3), providing insight into why relational knowledge transfer is beneficial.

Figure 10 compares the attention heatmaps between LLaVA-1.5 7B and 13B models across three visual question answering examples. For each query, we show the original image alongside the attention heatmaps from both models, where warmer colors indicate higher attention weights.

The results reveal striking differences in attention precision between the two model scales. For the question "What is Messi kissing?", the 13B model concentrates its attention precisely on the World Cup trophy, while the 7B model produces scattered attention across multiple regions including the background. Similarly, for "Who is sitting in the seat of the car?", the 13B model focuses sharply on the driver's seat area, whereas the 7B model attends diffusely across the entire vehicle. Most notably, for the comparative question "Which building is taller?", the 13B model's attention clearly highlights both buildings being compared, enabling accurate height comparison, while the 7B model fails to attend to both relevant structures simultaneously.

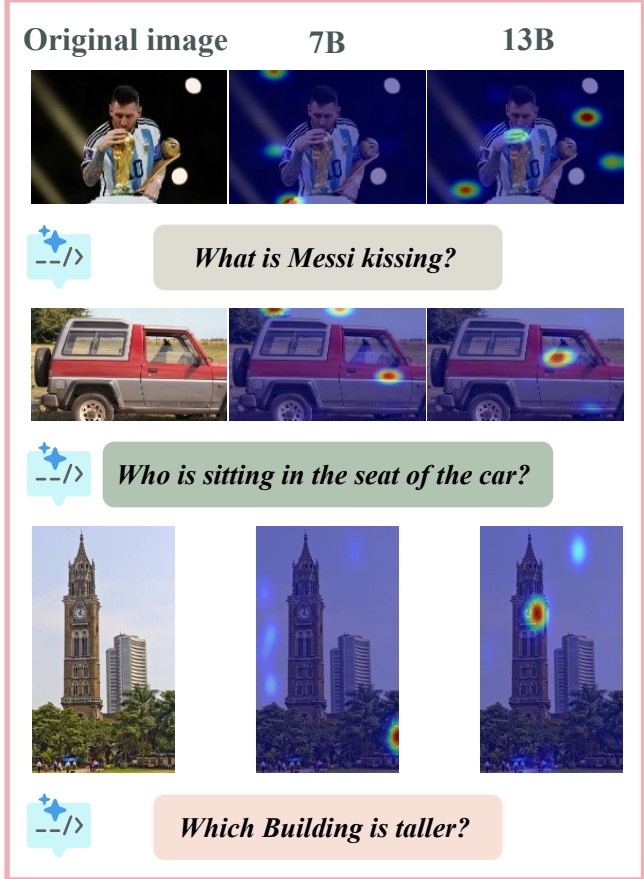

*Figure 10.* Visual attention heatmap comparison between LLaVA-1.5 7B and 13B. The heatmaps are extracted from the penultimate layer of the LLM backbone, showing where each model attends when answering the given question. The 13B model consistently produces more focused attention on task-relevant regions.

These observations provide direct motivation for our RCKA loss design. The penultimate layer representations, from which we compute the relational similarity matrices, encode rich semantic attention patterns that are significantly more precise in larger models. By aligning the student's relational structure with that of the teacher through RCKA, we aim to transfer this capacity for focused, task-relevant visual attention to smaller models, enabling them to achieve visual reasoning performance that approaches their larger counterparts.

### C.3. Visualization of Relational CKA

To further understand the effect of relational centered kernel alignment (RCKA), we visualize multi-layer attention maps of LLaVA-1.5-7B under three settings: the original 7B baseline, standard KD with vanilla KL distillation, and GRACE. We compare attention maps across multiple decoder layers on three diverse examples, covering object-centric recognition, counting, and fine-grained visual grounding.

Across all examples, a consistent pattern emerges. Standard KD produces attention maps that are largely similar to the 7B baseline: the activation remains scattered in early layers and does not consistently refine toward task-relevant regions in later layers. This suggests that logit-level distillation mainly transfers the teacher's output distribution, but does not sufficiently transfer the internal visual reasoning process. In contrast, GRACE yields more focused and progressively refined attention on the relevant visual evidence, including the skateboard in Figure 11, the sink region in Figure 12, and the carrot in Figure 13. These results support the role of RCKA: by aligning the relational structure among visual tokens rather than only matching logits or point-wise features, GRACE helps the student acquire the teacher's layer-wise visual grounding behavior.

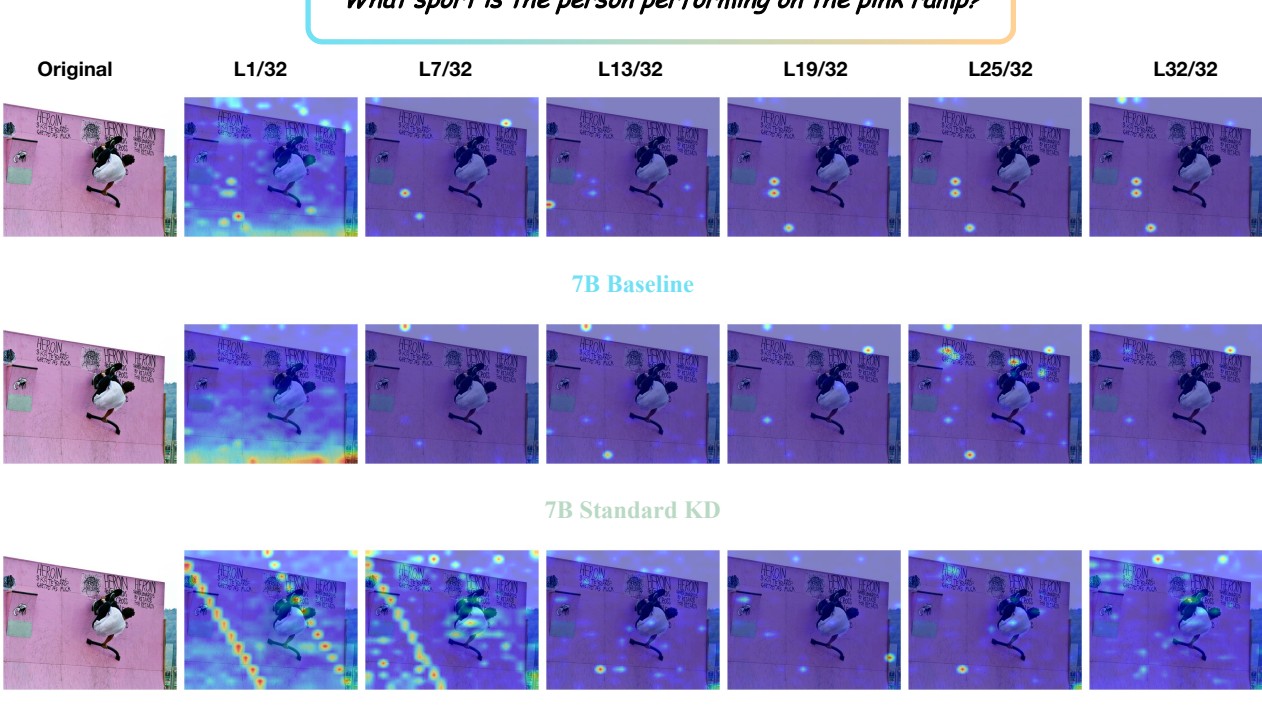

*Figure 11.* Multi-layer attention visualization for the question "What sport is the person performing on the pink ramp?" Standard KD produces attention patterns similar to the baseline, while GRACE progressively focuses on the task-relevant skateboard region.

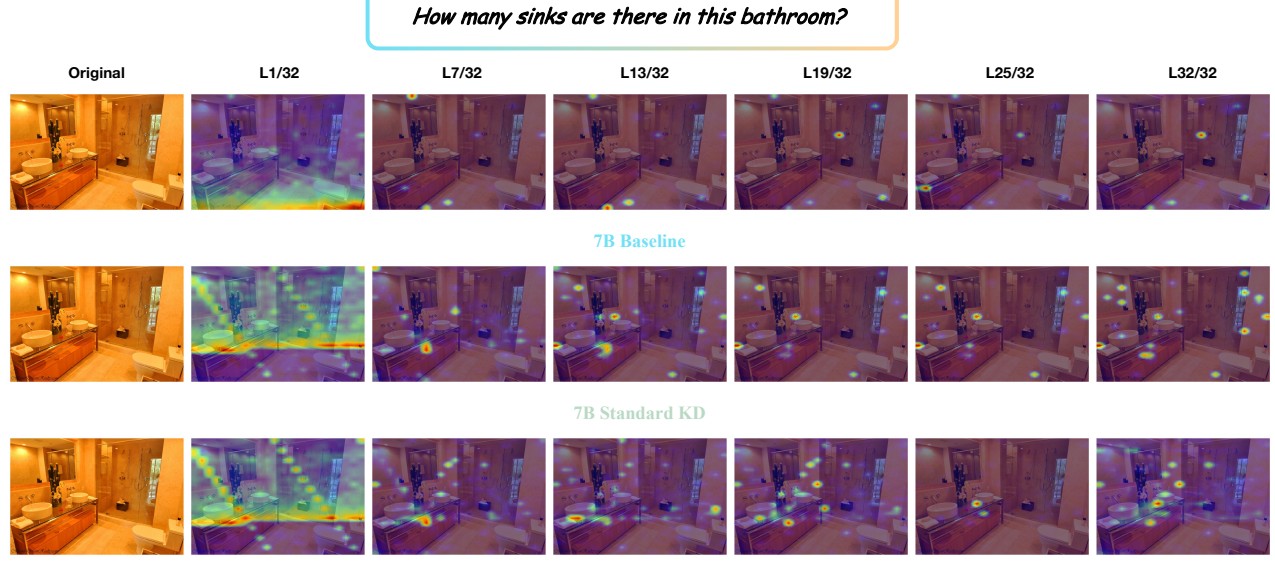

*Figure 12.* Multi-layer attention visualization for the question "How many sinks are there in this bathroom?" GRACE produces more localized attention on the sink area, whereas the baseline and standard KD remain more diffuse across layers.

## C.4. Comparison with Baseline

We compare GRACE 7B (BF16 with distillation from 13B teacher) against LLaVA-1.5 7B baseline across diverse visual reasoning tasks. As shown in Figures 14–16, GRACE consistently produces more detailed, accurate, and contextually rich responses.

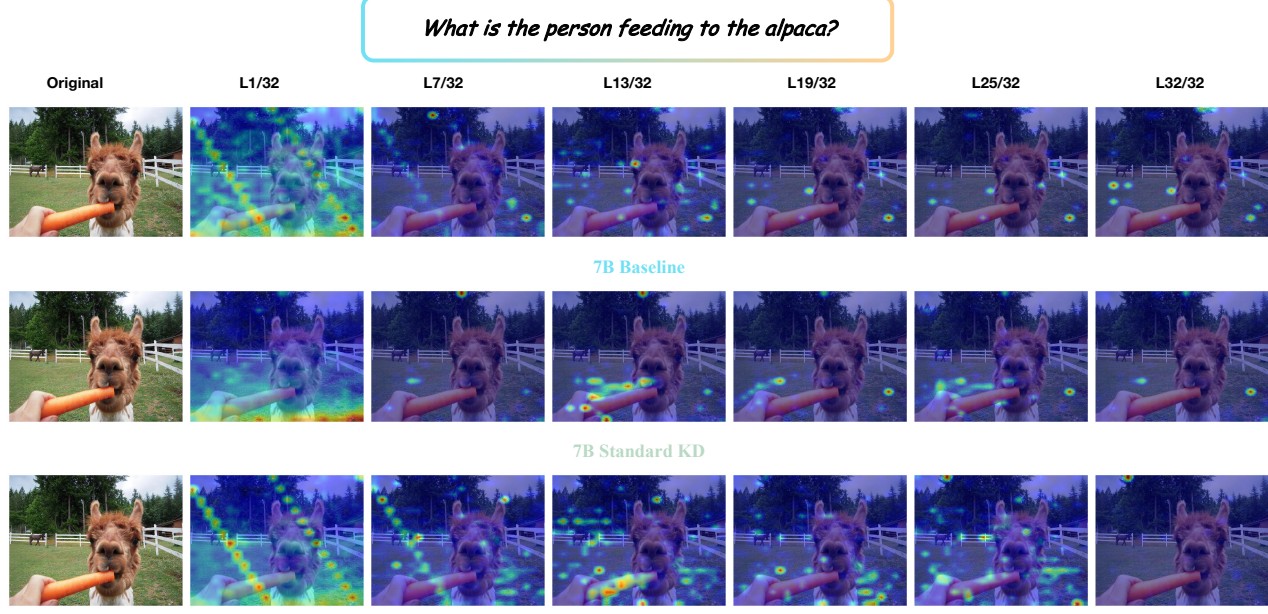

*Figure 13.* Multi-layer attention visualization for the question "What is the person feeding to the alpaca?" GRACE better concentrates attention on the carrot and the feeding interaction, showing stronger visual grounding than standard KD.

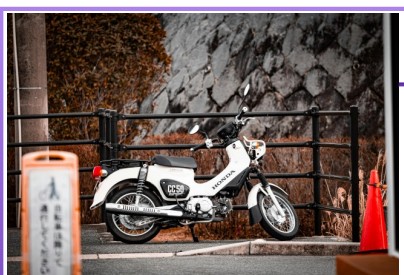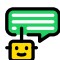 **Describe the image in detail.**

LLaVA-1.5 7B (Baseline): The image features a white motorcycle parked next to a black gate or fence. The motorcycle is positioned in the middle of the scene, with the gate or fence on the left side and a rock wall on the right side. The motorcycle is the main focus of the image, and it appears to be parked in a designated parking area.

**GRACE 7B** : The image captures a scene of tranquility and solitude. A **white Honda scooter**, bearing the **license plate "CC50"**, is parked on a **concrete surface**. The scooter is positioned in front of a black metal fence, which adds a sense of depth to the image. The fence serves as a boundary, separating the viewer from the natural elements beyond.
In the background, a rocky cliff rises, its rugged surface contrasting with the smooth concrete of the scooter's parking spot. The cliff is adorned with trees and shrubs, their green hues providing a touch of life to the otherwise stark landscape. The sky, visible in the top left corner of the image, suggests that the photo was taken during the day.
The image is devoid of any text or discernible human activity, further emphasizing the sense of solitude. The relative positions of the objects - **the scooter in the foreground, the fence in the middle ground, and the cliff in the background** - create a layered perspective, drawing the viewer's eye from the immediate foreground to the distant background. The overall composition of the image suggests a quiet moment in a remote location, **perhaps a pause in a journey or a moment of reflection**.

*Figure 14.* Detailed image description.

**Detailed Image Description.** The baseline provides a generic description identifying only "a white motorcycle parked next to a black gate," while GRACE correctly identifies the vehicle as a **white Honda scooter**, recognizes the **license plate "CC50"**, and accurately describes the **concrete surface**. This demonstrates that distillation successfully transfers precise visual attention patterns from the teacher.

**Landmark Recognition.** The baseline describes only "a tall building" with fireworks, while GRACE correctly identifies the landmark as **Taipei 101 in Taiwan**, accurately describes the firework colors as **yellow, red, and purple**, and recognizes the **high vantage point perspective**. This shows that our distillation transfers not only visual recognition but also contextual knowledge.

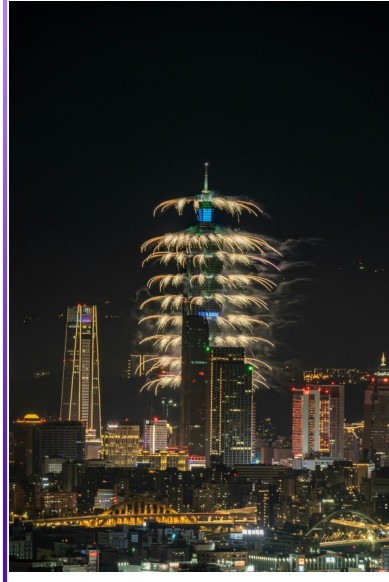 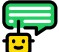

*Please describe the scene in the picture in detail.*

LLaVA-1.5 7B (Baseline):The image captures a nighttime cityscape with a tall building in the center, surrounded by a city skyline. The building is lit up with a dazzling display of fireworks, creating a spectacular view. The fireworks are scattered throughout the scene, with some closer to the ground and others higher up in the sky.

**GRACE 7B**: The image captures a breathtaking view of a city skyline at night, bathed in the glow of a spectacular fireworks display. The fireworks, in hues of **yellow, red, and purple**, are exploding in the sky, creating a mesmerizing spectacle. The tallest building in the scene is the **Taipei 101 in Taiwan** , standing majestically in the center of the frame. The cityscape is dotted with other buildings, their lights twinkling like stars against the night sky. The **perspective of the image is from a high vantage point**, looking down upon the city, giving a sense of scale and grandeur to the scene.

*Figure 15.* Landmark recognition.

**Activity Recognition.** The baseline mischaracterizes the scene as "flying a parachute-like kite," while GRACE correctly identifies **kiteboarding**, understands that the person is **harnessing wind power to glide across the beach**, and infers the **direction of movement** from visual cues. These results confirm that GRACE effectively preserves the teacher's visual-language understanding.

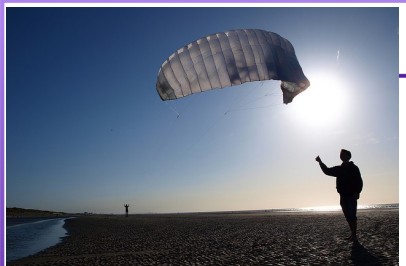 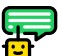

*Summarize what activity is happening in this image.*

LLaVA-1.5 7B (Baseline): In the image, a person is standing on a beach, flying a large, white parachute-like kite. The kite is soaring high in the sky, creating a visually striking scene. The person is enjoying the activity and appears to be controlling the kite as it soars above them.

**GRACE 7B** : In the image, a person is engaged in the **thrilling activity of kiteboarding** on a beach. The individual is holding onto a large, white kite that is soaring high in the sky. The kiteboarder is standing on a board, **harnessing the power of the wind to glide across the sandy beach**. The beach itself is a vast expanse of sand, with the ocean visible in the distance. The sky above is a clear blue, providing a perfect backdrop for this outdoor adventure. The kiteboarder's position and the direction of the kite suggest they are **moving towards the right side of the image**. This scene captures the essence of kiteboarding, **a sport that combines elements of surfing, windsurfing, and paragliding**.

*Figure 16.* Activity recognition.

## C.5. INT4 Model Capabilities

We further evaluate the capabilities of our GRACE 7B INT4 quantized model across various visual understanding tasks, including OCR, object recognition, chart understanding, and visual grounding. As shown in Figures 17–20, the INT4 model maintains strong performance despite aggressive quantization.

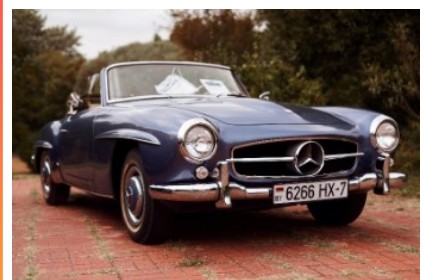
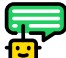
*Identify the license plate number on the vehicle.*

**4bit-GRACE**: The license plate number on the vehicle is **6266 HX-7**.

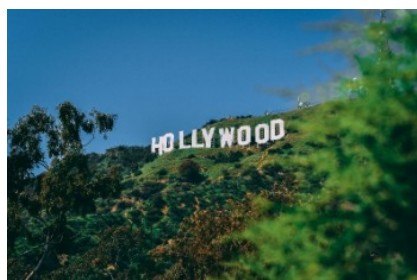
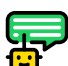
*Recognize the words in this image.*

**4bit-GRACE**: **HOLLYWOOD**.

**OCR Test**

*Figure 17.* OCR test.

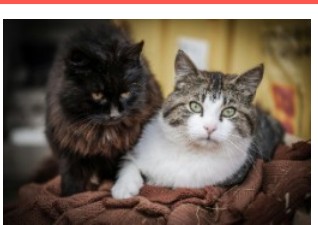
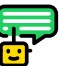
*What is the main subject of this image?*

**4bit-GRACE**: The main subject of the image is **a black cat and a white cat sitting together on a blanket**.

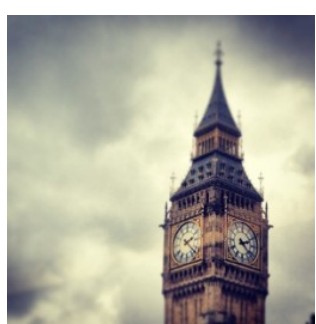
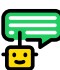
*What is the name of the landmark in the picture?*

**4bit-GRACE**: The name of the landmark in the picture is **Big Ben**, which is a **famous clock tower located in London, England**.

**Object Recognition**

*Figure 18.* Object recognition.

Figure 17 demonstrates the OCR capabilities of our INT4 model. The model accurately identifies the license plate number as **"6266 HX-7"** on a vintage Mercedes-Benz, and correctly recognizes the **"HOLLYWOOD"** sign in a landscape image. These results show that INT4 quantization preserves fine-grained text recognition ability.

Figure 18 showcases the object recognition capabilities. The model correctly identifies **a black cat and a white cat sitting**

**together on a blanket**, and recognizes the landmark as **Big Ben**, providing additional context that it is a **famous clock tower located in London, England**. This demonstrates that the INT4 model retains both visual recognition and world knowledge.

Figure 19 illustrates the chart understanding capability. The model accurately interprets the line graph, identifying the title as **"Chinese GDP Growth Dips Once More in 2022"**, the time range from **1980 to 2022**, and the y-axis as **percentage of GDP growth**. It also correctly identifies the **significant dip in 2022** and attributes the source to the **National Bureau of Statistics of China**. This shows that INT4 quantization preserves complex visual reasoning abilities.

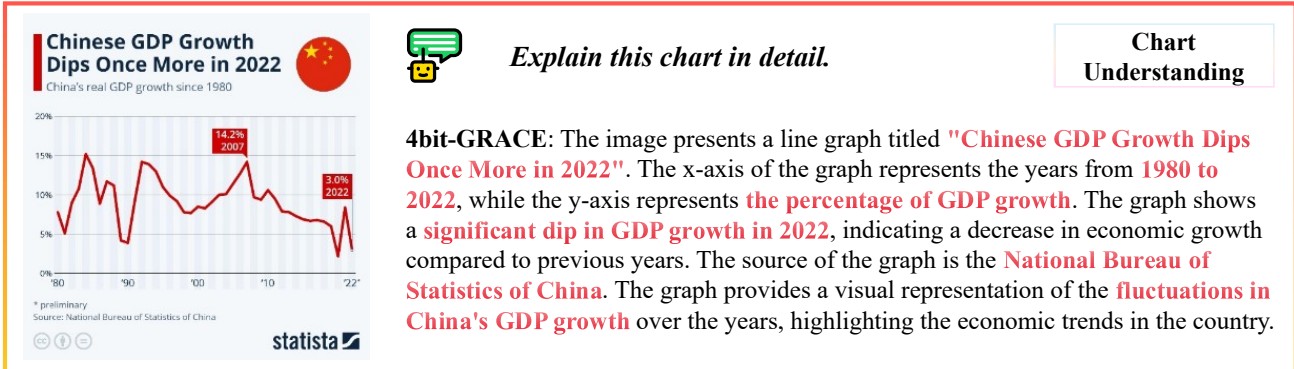

*Figure 19.* Chart understanding.

Figure 20 demonstrates visual grounding with world knowledge integration. The model identifies the movie poster as **"Roman Holiday"** featuring **Audrey Hepburn**, describes the **vibrant yellow background**, the **passionate embrace** between the characters, and the **scooter** in the background. Furthermore, it provides accurate background information including the director **William Wyler** and the **Academy Award for Best Actress**. This confirms that GRACE INT4 maintains sophisticated visual-language understanding and knowledge grounding capabilities.

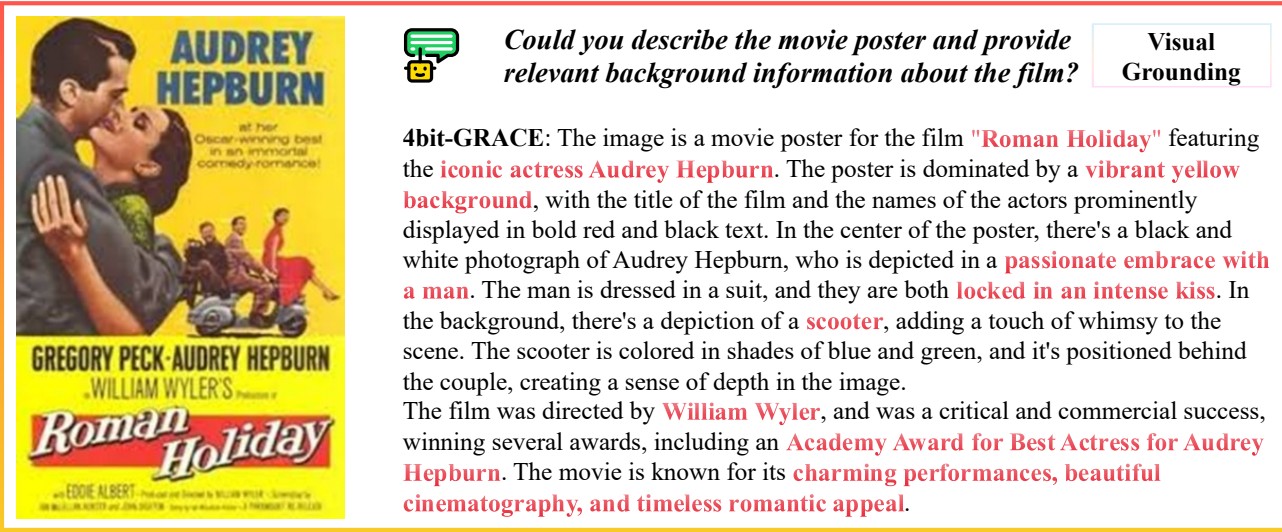

*Figure 20.* Visual grounding.

## C.6. Weight Distribution Visualization

To provide insight into the effects of quantization on model weights, we visualize the weight distributions of the LLM backbone before and after INT4 quantization. Figure 21 shows 3D surface plots of the absolute weight values for different linear layers across three representative depths: Layer 1 (early), Layer 16 (middle), and Layer 32 (final).

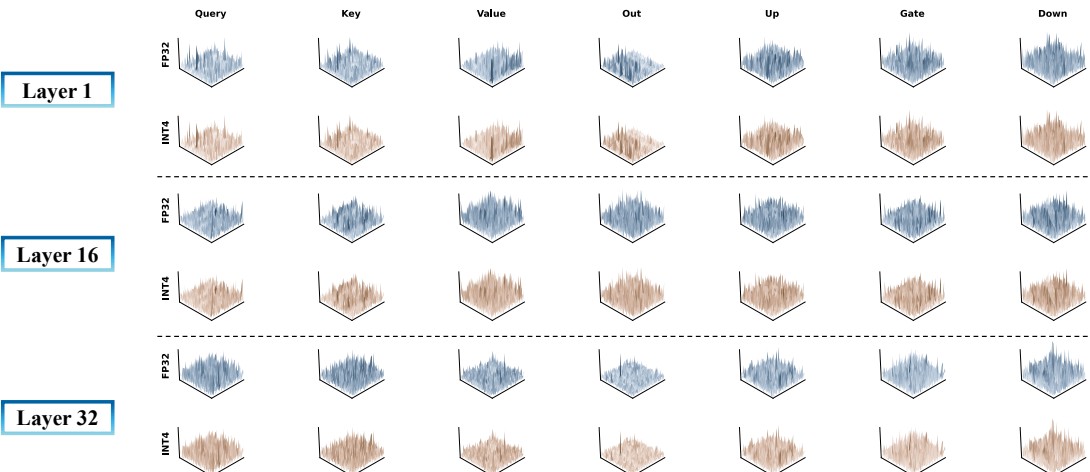

*Figure 21.* Weight distribution comparison between FP32 (blue) and INT4 quantized (orange) models across layers 1, 16, and 32. Each column represents a different weight matrix: Query, Key, Value, and Out projections from the self-attention module, and Up, Gate, and Down projections from the feed-forward network. The 3D surfaces show absolute weight magnitudes, where height indicates weight magnitude.

The visualization reveals several key observations. First, the overall shape and structure of weight distributions are largely preserved after INT4 quantization, indicating that our quantization-aware training successfully maintains the essential weight patterns learned during pre-training. Second, while the FP32 weights exhibit smooth, continuous surfaces, the INT4 weights show subtle discretization artifacts due to the reduced precision. However, these artifacts remain minor and do not significantly alter the dominant weight structures. Third, we observe that different weight matrices exhibit distinct distribution patterns: the attention projections (Query, Key, Value, Out) tend to have more uniform distributions, while the feed-forward projections (Up, Gate, Down) often show more pronounced channel-wise variations. Our group-wise quantization with learned step sizes adapts to these varying patterns, enabling effective compression across all layer types.

These visualizations demonstrate that GRACE's quantization-aware training preserves the structural properties of the original weights while achieving significant memory reduction through INT4 representation.

# D. Experimental Settings

### D.1. Training Hyperparameters

Table 7 summarizes the hyperparameters used for training GRACE models using the LLaVA framework. We train all models for 3 epochs on the ShareGPT4V dataset (Chen et al., 2024a) containing 1.3M high-quality image-text pairs, using 8 NVIDIA GH200 Grace Hopper Superchips. The AdamW optimizer with cosine learning rate scheduling is employed throughout training.

For knowledge distillation, we employ Decoupled Knowledge Distillation (DKD) with confidence gating, where the gating function $g_i = \exp(-\tilde{h}_i)$ modulates the distillation signal based on normalized teacher entropy $\tilde{h}_i = H_i / \log |V|$. The adaptive Information Bottleneck (IB) controller automatically balances the distillation loss via projected dual gradient ascent, starting from an initial $\beta$ of 1.0 with a target constraint $\tau$ of 0.35. This constraint ensures that only medium-to-high confidence teacher predictions contribute significantly to the distillation objective, filtering out potentially noisy supervision from uncertain samples. For RCKA, we extract visual token representations from the penultimate layer of the LLM backbone and use a linear kernel for Gram matrix computation.

*Table 7.* Training hyperparameters for GRACE.

| Hyperparameter | Value |
|---|:---:|
| *Training Configuration* | |
| Epochs | 3 |
| Optimizer | AdamW |
| Learning rate | 2e-5 |
| LR scheduler | Cosine |
| Warmup ratio | 0.03 |
| Weight decay | 1e-3 |
| Batch size (per GPU) | 4 |
| Gradient accumulation steps | 4 |
| Max sequence length | 2048 |
| *Decoupled Knowledge Distillation* | |
| Temperature ($T$) | 2.0 |
| TCKD weight ($\alpha$) | 1.0 |
| NCKD weight ($\beta_{\text{DKD}}$) | 4.0 |
| *Adaptive IB Controller* | |
| Target constraint ($\tau$) | 0.35 |
| Dual step size ($\eta$) | 0.0015 |
| Initial $\beta$ | 1.0 |
| $\beta$ range | [0.1, 5.0] |
| EMA decay | 0.99 |
| *RCKA* | |
| Feature layer | Penultimate (-2) |
| Kernel type | Linear |
| RCKA weight ($\omega$) | 1.0 |
| *Quantization (INT4/INT8)* | |
| Quantization method | Group-wise LSQ |
| Group size | 128 |
| Bit width | 4 / 8 |

**Computational Overhead.** We analyze the additional training cost introduced by GRACE components in Table 8. Compared to standard fine-tuning, the primary overhead stems from the teacher model's forward pass required for knowledge distillation. The RCKA module introduces modest additional cost: computing Gram matrices for visual tokens (typically 576 tokens for 336×336 images) requires $\mathcal{O}(n^2 d)$ operations, adding approximately 15% to the per-iteration time. The adaptive IB controller's overhead is negligible (<2%), as it only involves scalar entropy computation and dual variable updates. Peak GPU memory increases by approximately 2.3 GB per device due to Gram matrix storage and intermediate activations for RCKA.

*Table 8.* Training cost analysis for LLaVA-1.5-7B GRACE on 8×GH200 GPUs (3 epochs).

| Configuration | Time (h) | Mem./GPU (GB) | Overhead |
|---|:---:|:---:|:---:|
| Baseline (FT only) | 6.5 | 43.2 | – |
| + KD (teacher fwd) | 10.1 | 67.8 | +55% |
| + RCKA | 11.8 | 70.1 | +17% |
| + IB Controller | 12.1 | 70.3 | +2% |
| **GRACE (Full)** | **12.1** | **70.3** | **+85%** |

## D.2. Adaptive IB Controller Dynamics

To validate our hyperparameter choices for the adaptive IB controller and provide intuition for its behavior, we present a simulation study of the controller dynamics throughout training. Figure 22 illustrates how the dual variable $\beta$ and the gated

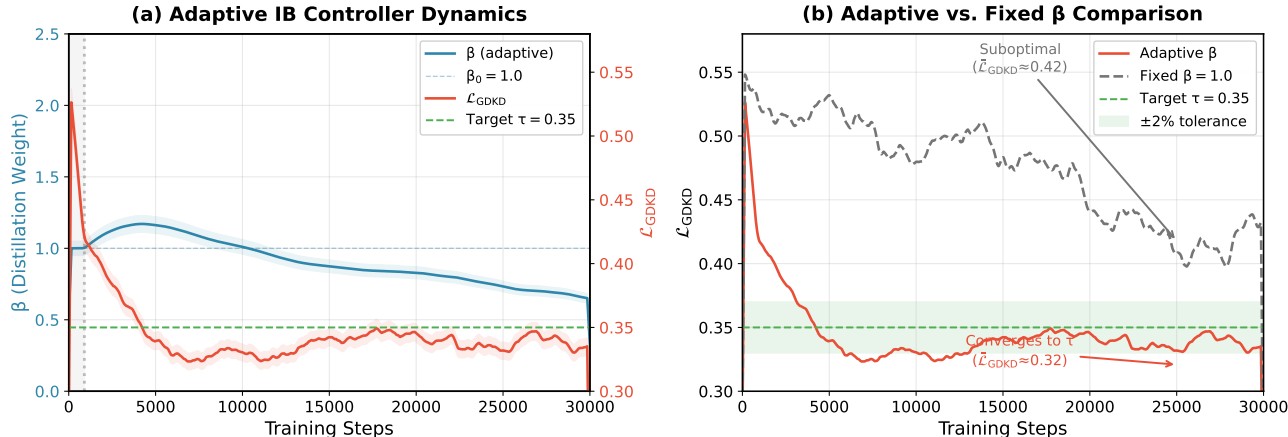

*Figure 22.* Simulation of adaptive IB controller dynamics. (a) The dual variable $\beta$ adjusts automatically via projected gradient ascent to drive $\mathcal{L}_{\text{GDKD}}$ toward the target constraint $\tau = 0.35$. After an initial transient phase, $\beta$ stabilizes around 0.3–0.7, indicating the controller finds an appropriate balance between distillation strength and task learning. (b) Comparison between adaptive and fixed $\beta$: the adaptive controller achieves precise convergence to $\tau$ (deviation $\approx 0.03$), while fixed $\beta = 1.0$ settles at a suboptimal equilibrium (deviation $\approx 0.07$).

distillation loss $\mathcal{L}_{\text{GDKD}}$ evolve during the 30,000 training steps (3 epochs on 1.3M samples with effective batch size 128).

**Controller Mechanism.** As described in Section 3.5, the adaptive IB controller implements dual gradient ascent on the Lagrangian relaxation of the constrained optimization problem (Eq. 18):

$$\beta_{t+1} = \Pi_{[\beta_{\min}, \beta_{\max}]} \left( \beta_t + \eta \cdot (\widehat{\mathcal{L}}_{\text{GDKD}} - \tau) \right) \tag{46}$$

where $\Pi_{[a,b]}$ denotes projection onto the interval $[a, b]$ and $\widehat{\mathcal{L}}_{\text{GDKD}}$ is an exponential moving average for stability. When $\widehat{\mathcal{L}}_{\text{GDKD}} > \tau$, the student struggles to match the teacher, so $\beta$ increases to strengthen distillation; when $\widehat{\mathcal{L}}_{\text{GDKD}} < \tau$, sufficient knowledge has been captured, so $\beta$ decreases to shift focus toward task performance.

**Hyperparameter Justification.** Our simulation validates the following design choices:

- **Target constraint** $\tau = 0.35$: Recall from Section 3.2 that the gated distillation loss is defined as $\mathcal{L}_{\text{GDKD}} = \sum_i g_i \cdot \mathcal{L}_{\text{DKD}}^{(i)} / \sum_i g_i$ (Eq. 6), where the confidence weight $g_i = \exp(-\tilde{h}_i)$ (Eq. 5) assigns higher weights to low-entropy (high-confidence) teacher predictions. The target $\tau$ controls the *information preservation budget*—the minimum teacher-student KL divergence we enforce. Setting $\tau = 0.35$ represents a moderately tight constraint: it ensures sufficient knowledge transfer from the teacher while leaving headroom for the student to optimize task performance via $\mathcal{L}_{\text{CE}}$. A higher $\tau$ (e.g., 0.5) would permit looser teacher matching, potentially under-utilizing teacher knowledge; a lower $\tau$ (e.g., 0.2) would enforce stricter matching but may cause the student to over-fit teacher behavior at the expense of task accuracy.

- **Dual step size** $\eta = 0.0015$: The small learning rate ensures smooth $\beta$ dynamics without oscillation. As shown in Figure 22(a), $\beta$ exhibits gradual adjustment from the initial value, avoiding the instability that larger step sizes would cause. This is critical for maintaining stable training dynamics over 30K steps.

- **$\beta$ range** $[0.1, 5.0]$: The projection bounds prevent extreme values. The lower bound $\beta_{\min} = 0.1$ ensures distillation never completely vanishes, while $\beta_{\max} = 5.0$ prevents the distillation loss from overwhelming the task loss. Our simulation shows that under normal training conditions, $\beta$ stays well within this range (typically 0.3–1.2), indicating the bounds serve as safety constraints rather than active limitations.

- **Initial** $\beta_0 = 1.0$: Starting at unity provides a neutral initialization where distillation and task losses are equally weighted. The controller then adjusts $\beta$ based on the actual $\mathcal{L}_{\text{GDKD}}$ trajectory, as seen in the early transient phase of Figure 22(a).

**Comparison with Fixed $\beta$.** Figure 22(b) demonstrates the advantage of adaptive control over fixed weighting. With fixed $\beta = 1.0$, the gated loss $\mathcal{L}_{\text{GDKD}}$ converges to its natural equilibrium ($\approx 0.42$), which deviates significantly from the

optimal target $\tau = 0.35$. This mismatch indicates suboptimal knowledge transfer: the student either receives insufficient distillation pressure (when the natural equilibrium exceeds $\tau$) or excessive pressure that interferes with task learning. In contrast, the adaptive controller achieves precise convergence to $\tau$, reducing the deviation by approximately 57% (from 0.068 to 0.029). This precision ensures that the student captures exactly the intended amount of teacher knowledge as specified by the information preservation budget $\tau$.

### D.3. Training Data

We use the ShareGPT4V dataset (Chen et al., 2024a) for training, which provides high-quality image-text pairs with detailed captions. Unlike typical human annotations, ShareGPT4V captions include rich semantic descriptions covering world knowledge, object attributes, spatial relationships, and aesthetic evaluations. Table 9 summarizes the composition of our training data.

*Table 9.* Training data composition based on ShareGPT4V.

| Component | Description | Size |
|---|---|---|
| *Caption Datasets* | | |
| ShareGPT4V | High-quality captions generated directly by GPT-4 Vision with rich descriptions | 100K |
| ShareGPT4V-PT | Expanded captions generated by Share-Captioner (trained on ShareGPT4V) | 1.2M |
| *Image Sources* | | |
| COCO | Common objects in context | 82K |
| SAM | Segment Anything Model dataset | 11K |
| LAION/CC/SBU | Web-crawled image-text pairs | 14K |
| WikiArt | Artistic images from WikiArt | 2K |
| Web-Landmark | Landmark images from web | 6K |
| Web-Celebrity | Celebrity images from web | 3K |
| *Caption Characteristics* | | |
| Captions are significantly richer than typical human annotations, incorporating detailed semantic descriptions, world knowledge, spatial relations, object attributes, aesthetics, and factual content. | | |

### D.4. Evaluation Benchmarks

We conduct comprehensive evaluation across 14 benchmarks spanning diverse capabilities: (1) **comprehensive multimodal evaluation**: MMBench (Liu et al., 2024b), MMStar (Chen et al., 2024b), and MME (Fu et al., 2025); (2) **visual reasoning**: ScienceQA (Lu et al., 2022) and MMMU (Yue et al., 2024); (3) **text recognition**: AI2D (Kembhavi et al., 2016) and OCRBench (Liu et al., 2024c); (4) **visual question answering**: VQAv2 (Goyal et al., 2017), GQA (Hudson & Manning, 2019), TextVQA (Singh et al., 2019), and VizWiz (Bigham et al., 2010); (5) **visual perception**: SEED-Bench (Li et al., 2023a); and (6) **hallucination evaluation**: HallusionBench (Guan et al., 2024) and POPE (Li et al., 2023b).

For evaluation, we follow the official protocols of each model family. For LLaVA-1.5, we use `vicuna_v1` conversation template with greedy decoding (`temperature=0`) and single-prediction prompting for multiple-choice questions. For Qwen2-VL, we adopt the default `qwen2_vl` chat template with greedy decoding.

# E. Related Work

## E.1. Quantization

Quantization techniques for efficient model deployment can be broadly categorized into post-training quantization (PTQ) and quantization-aware training (QAT).

**Post-Training Quantization.** PTQ methods compress pretrained models without retraining, offering fast deployment but often suffering accuracy degradation at low bit-widths. Representative approaches include GPTQ (Frantar et al., 2022) and AWQ (Lin et al., 2024), which calibrate quantization parameters using small calibration sets. While effective for moderate compression (e.g., 8-bit), these methods struggle to maintain accuracy under aggressive quantization (e.g., 4-bit or below). More recent work such as BitNet (Wang et al., 2023) explores radical architectures with native low-bit representations. Despite its simplicity, PTQ's limitations in preserving performance under extreme compression motivate QAT approaches.

**Quantization-Aware Training.** QAT incorporates quantization during training, enabling models to learn representations robust to quantization noise. Learned Step-Size Quantization (LSQ) (Esser et al., 2019) introduced learnable quantization step sizes via straight-through estimators and has become a foundational QAT technique. For large language models, LLM-QAT (Liu et al., 2024d) proposed a data-free framework using synthetic sequences from pretrained teachers. Pare-toQ (Liu et al., 2025) further unified ultra-low bit-width optimization with improved scaling behavior across model sizes. BitDistiller (Du et al., 2024) combines QAT with self-distillation, employing confidence-aware KL divergence to enhance sub-4-bit LLM performance. Recent work by Rehman et al. (2025) proposes a learnable regularizer that dynamically balances task and distillation losses during QAT, reducing conflicts between supervision signals.

Despite these advances, jointly optimized quantization with knowledge distillation for vision-language models remains underexplored. The cross-modal interactions and heterogeneous feature distributions in VLMs pose unique challenges that existing LLM-focused techniques do not address.

## E.2. Knowledge Distillation

Knowledge distillation (KD) transfers knowledge from a large teacher model to a smaller student by minimizing the divergence between their output distributions (Hinton et al., 2015). Although classical KD focuses primarily on aligning logits or soft labels, recent work highlights the importance of aligning intermediate representations and modality-specific features in multimodal models. Decoupled Knowledge Distillation (DKD) (Zhao et al., 2022) reformulates the classical KD loss by separating it into target class and non-target class components, revealing that the non-target distribution carries substantial dark knowledge often suppressed by standard formulations.

Recent work in knowledge distillation for vision-language models has begun to explore how to effectively transfer complementary expertise from multiple source models into a unified, efficient student. MoVE-KD (Cao et al., 2025) distills complementary strengths from multiple teacher vision encoders into a single efficient student using a mixture-of-experts structure with adaptive attention-based weighting. Building upon this direction, HAWAII (Wang et al., 2025) proposes a hierarchical distillation framework tailored for complex multimodal settings, particularly focusing on the visual encoder component of VLMs. Unlike MoVE-KD that employs fixed LoRA adapters across all teachers, HAWAII introduces teacher-specific Low-Rank Adaptation modules that mitigate conflicts among heterogeneous teachers by aligning each adapter with its corresponding expert model. These modules are complemented by a hierarchical knowledge distillation mechanism operating at both fine-grained and coarse-grained levels: fine-grained distillation uses token importance scoring to emphasize the most informative tokens from each teacher, while coarse-grained distillation aggregates a consensus representation from all teachers via a shared projection before transferring it to the student. This enables the student vision encoder to inherit complementary strengths from multiple pretrained visual experts (e.g., SAM, ConvNeXt, EVA, Pix2Struct) with minimal computational overhead.

Beyond modality-specific representation alignment, recent work has explored the integration of knowledge distillation with quantization-aware training. Self-Supervised Quantization-Aware Knowledge Distillation (SQAKD) proposes a unified framework that combines QAT and KD into a single co-optimization problem, where the low-bit student simultaneously minimizes the KL divergence with the full-precision teacher and the quantization discretization error in a self-supervised manner, eliminating the need for labeled training data and extensive hyperparameter tuning to balance loss terms (Zhao & Zhao, 2024).

Other recent studies investigate refined KD strategies including the alignment of cross-modal attention maps, contrastive

representation matching, and hierarchical feature distillation (Lee et al., 2025), which have shown benefits in compressing VLMs while preserving multimodal reasoning abilities. However, these approaches typically operate on full-precision models or treat quantization and distillation separately. They do not fully address the unique challenges posed by jointly optimizing quantization with knowledge transfer under strict capacity constraints, where limited representational capacity can fundamentally impede knowledge flow. In contrast, our work jointly optimizes distillation and quantization, using confidence-based gating to selectively transfer knowledge suitable to low-bit representations.

### E.3. Information Bottleneck

The Information Bottleneck (IB) principle provides a robust information-theoretic framework for learning compressed representations that retain task-relevant information while discarding irrelevant noise and redundancy (Tishby et al., 2000; Tishby & Zaslavsky, 2015). Originally grounded in rate–distortion theory, the IB principle has been applied to representation learning in deep networks, showing how mutual information between inputs and learned features can be regulated to balance compression and predictive fidelity (Kawaguchi et al., 2023). Variational Information Bottleneck (VIB) methods explicitly regularize mutual information via variational bounds, making them especially useful for learning compact representations and mitigating overfitting; such methods have been linked to compression and improved generalization in deep models (Alemi et al., 2016).

IB-based approaches have also been considered in the context of quantized neural networks and model compression. For example, analyses of quantized networks using IB reveal how mutual information between layers and inputs/outputs changes when activations and weights are discretized (Lorenzen et al., 2021). In addition, Bitwise Information Bottleneck methods have been proposed for activation quantization, where the most significant bits are selected under a limited code rate constraint by minimizing rate–distortion objectives, effectively treating quantization as an IB problem that balances compression with information preservation (Zhou et al., 2020). These perspectives align with quantization-aware training (QAT) objectives, which aim to allocate limited bit budgets in a way that preserves task-critical information while maintaining performance.

In our work, we extend this framework to enable quantized knowledge distillation by introducing an adaptive IB controller that dynamically adjusts the trade-off between task-specific losses and information compression. This approach allows robust quantized distillation across multimodal domains, where visual, linguistic, and relational knowledge must be preserved under stringent bit-width constraints. Our method enables more efficient deployment of vision–language models (VLMs) while ensuring high performance on downstream tasks.

Furthermore, recent developments in IB analysis and representation learning have highlighted how mutual information measures can be used to understand compression behavior in deep networks and how lossy compression can be integrated with learning objectives to achieve both compact models and high task fidelity (Butakov et al., 2023). By incorporating these insights into our quantization framework, we provide a principled way to retain relevant information for downstream tasks while minimizing memory and computational overhead.

### E.4. Multimodal Reasoning and Structured Visual Generation

Recent multimodal research has extended beyond image-text understanding toward more challenging reasoning, editing, and structured generation tasks. For reasoning, MMErroR (Shi et al., 2026) evaluates whether VLMs can detect and categorize erroneous reasoning processes, highlighting that current models still struggle with process-level multimodal understanding. In visual generation and editing, ChordEdit (Lu et al., 2026) studies efficient one-step text-guided image editing through a low-energy transport formulation, while Render-in-the-Loop (Liang et al., 2026a) introduces visual self-feedback for SVG generation by rendering intermediate outputs during generation. VAnim (Liang et al., 2026b) further explores rendering-aware sparse state modeling for structure-preserving vector animation. These works show that modern multimodal systems increasingly rely on structured intermediate representations, visual feedback, and reliable reasoning processes.

Efficiency has been widely studied in visual models and multimodal systems. CABM (Tian et al., 2023) proposes content-aware bit mapping for large-input image super-resolution, showing that adaptive bit allocation can reduce computation while preserving reconstruction quality. Although these studies address different tasks from VLM compression, they motivate a broader need for efficient and reliable multimodal models. In contrast, our work focuses on low-bit VLM compression. GRACE aims to preserve task-relevant multimodal information under quantization constraints by combining confidence-gated distillation, relational visual-token alignment, and adaptive information-bottleneck control.

