# OpenReview forum: "Gated Relational Alignment via Confidence-based Distillation for Efficient VLMs"
_ICML.cc/2026/Conference — ICML 2026 regular_

### Official Review · Reviewer_FKph · 2026-02-27

**Soundness:** 2
**Presentation:** 3
**Significance:** 3
**Originality:** 2
**Overall Recommendation:** 4
**Confidence:** 2

**Summary:**

The paper introduces GRACE (Gated Relational Alignment via Confidence-based Distillation for Efficient VLMs), a framework designed to compress Vision-Language Models (VLMs) through a combination of Quantization-Aware Training (QAT) and Knowledge Distillation (KD). The authors frame this joint optimization under the Information Bottleneck (IB) principle, where quantization acts as a capacity constraint and distillation determines what information is preserved. The framework consists of three main components: (1) confidence-gated decoupled knowledge distillation (GDKD) to filter noisy teacher supervision, (2) relational-centered kernel alignment (RCKA) to transfer intra-sample visual token structures, and (3) an adaptive IB controller to balance the distillation and task losses. Extensive experiments on LLaVA-1.5 and Qwen2-VL show that the proposed INT4 models achieve strong performance, often surpassing the original undistilled BF16 baselines.

**Compliance With Llm Reviewing Policy:**

Affirmed.

**Final Justification:**

Most of my concerns have been addressed.

**Key Questions For Authors:**

See Weaknesses. If my concerns can be addressed, I will improve my score.

**Limitations:**

yes

**Strengths And Weaknesses:**

Strengths：

1、Principled Theoretical Framing: The formulation of VLM compression through the lens of the Information Bottleneck principle is theoretically elegant. Treating quantization as a hard capacity constraint while using the teacher's output to maximize task-relevant information preservation provides a solid foundation for the joint optimization of QAT and KD.

2、Strong Empirical Results: The experimental results are robust across multiple architectures (LLaVA-1.5 and Qwen2-VL). The ability of an INT4 model to recover and even slightly exceed the performance of the original FP16 student baseline (e.g., LLaVA-1.5 4-bit GRACE achieving 67.2% vs. the FP16 baseline of 66.5% ) is practically highly valuable for edge deployment.

Weaknesses：

1、The motivational claim regarding Figure 3 is incomplete. The authors state that logit-based distillation cannot transfer semantic attention patterns, pointing to Figure 3 to show that the 7B student fails to localize the object compared to the 13B teacher. However, Figure 3 only illustrates the attention gap between the teacher and a vanilla (pre-distillation) student. To conclusively claim that logit-based KD is insufficient, the authors must show the attention maps of a 7B model trained explicitly with standard logit KD. Without this, the visual necessity of the RCKA module is not perfectly justified, as it remains unproven that standard KD fails to correct these attention maps.

2、The adoption of Decoupled Knowledge Distillation (DKD) lacks specific motivation within the context of this paper. While the proposed confidence gating mechanism elegantly addresses the issue of unreliable teacher supervision , the use of DKD itself appears to be a plug-and-play addition from prior work. The paper does not provide ablation studies showing why standard KL divergence combined with confidence gating wouldn't be just as effective. The authors need to better justify why decoupling the target and non-target classes is mathematically or empirically necessary for this specific VLM quantization task.

3、The specific necessity of QAT versus advanced PTQ pipelines is not fully isolated. While the authors compare against standard PTQ methods (like AWQ, RTN) in Table 2, it would be more convincing to see a comparative pipeline where the model is first distilled in FP16 using GRACE, and then quantized using an advanced PTQ method. This would isolate whether the joint optimization (QAT + KD) is truly the source of the performance retention, or if the GRACE KD recipe simply produces a more robust FP16 model that can survive standard PTQ.

---

> ### Author Rebuttal · Authors · 2026-03-27
>
> Dear reviewer,
>
> we sincerely thank you for the thorough and constructive evaluation. The questions precisely identified the gaps in our empirical justification, and the suggested experiments have significantly strengthened our work. We address each point below with new experiments and analysis.
>
> # W1
>
> We use the LLaVA-1.5 7B model with standard KD (vanilla KL) and extract its attention maps for direct comparison. **Fig. 5(a)–(c) in this link** (https://anonymous.4open.science/r/grace_rebuttal-56F0) visualize multi-layer attention heatmaps across **7B Baseline**, **7B Standard KD**, and **7B GRACE** on three diverse examples.
>
> Across all cases, a consistent pattern emerges:
>
> - **7B Standard KD** produces attention maps largely indistinguishable from the baseline. Activation remains scattered and diffuse across all layers, with no progressive localization toward task-relevant regions.
> - **7B GRACE** shows markedly stronger and more focused attention on task-relevant regions (skateboard, sink, carrot), with clear progressive refinement from early to late layers that mirrors the teacher's behavior.
>
> These results validate the necessity of RCKA: logit-based KD transfers *what* the teacher predicts but not *how* it attends. RCKA explicitly aligns the relational structure among visual tokens (Eq. 10–13), enabling the student to acquire the teacher's progressive attention refinement capability.
>
> ___
>
> # W2
>
> **Two properties of VLMs make decoupling especially critical for quantization**, beyond general KD settings: (1) Large vocabularies ($|V| \approx$ 32K–152K) cause the gradient w.r.t. a non-target logit $z_i$ to be bounded by $p_i^T \to 0$, leading to vanishing gradients for non-target classes under standard KL. DKD solves this by renormalizing the non-target distribution $\hat{P}^{nt}$ (Eq. 2), making gradient magnitude independent of the target-class probability. This signal is further amplified by setting $\beta_{\text{dkd}} = 4.0 > \alpha = 1.0$ (Table 6). (2) Low-bit quantization disrupts the subtle ordering among non-target logits where margins are small, while the target class retains its large margin [1,2]. Decoupling enables independent preservation of this fragile ranking structure via up-weighted NCKD.
>
> **New ablation** on Qwen2-VL (7B→2B), BF16, **without** RCKA or Adaptive IB, to purely isolate the logit-level distillation formulation:
>
> | Method | MMMU | DocVQA | MathVista | Avg |
> |:-------|:----:|:------:|:---------:|:---:|
> | (a) Standard KL | 47.9 | 89.1 | 46.2 | 61.1 |
> | (b) Standard KL + Gating | 48.2 | 89.3 | 46.7 | 61.4 |
> | (c) DKD (no gating) | 49.4 | 89.5 | 48.0 | 62.3 |
> | **(d) GDKD (ours)** | **50.1** | **89.8** | **49.3** | **63.1** |
>
> **Key findings:**
>
> - **With gating held fixed, decoupling adds +1.7 in average (b→d)**.
>
> - The gain is **task-dependent**: MathVista shows the largest improvement (**+2.6**), as mathematical reasoning relies on understanding *which wrong answers are numerically closer to correct*, which is  exactly the relational ranking among candidates that NCKD captures. DocVQA shows a smaller gain (+0.5), as extractive QA produces peaked teacher distributions with limited dark knowledge.
>
> - Gating and DKD exhibit a **super-additive interaction**: gating alone contributes +0.3, DKD alone +1.2, summing to +1.5. But their combination yields **+2.0** over the standard KL baseline. This confirms that gating helps DKD by filtering noisy supervision *before* amplifying the non-target distribution.
>
> ___
>
> # W3
>
> We apply GPTQ-Int4 and AWQ-Int4 to our GRACE-distilled BF16 model, comparing against our joint QAT+KD pipeline. All experiments use Qwen2-VL (7B→2B).
>
> | Pipeline | MMMU | DocVQA | MathVista | Avg |
> |:---------|:----:|:------:|:---------:|:---:|
> | BF16 Baseline | 45.4 | 88.3 | 44.4 | 59.4 |
> | BF16 → GPTQ-Int4 | 36.2 | 87.2 | 41.7 | 55.0 |
> | BF16 → AWQ-Int4 | 37.3 | 87.0 | 39.9 | 54.7 |
> | GRACE BF16 (KD only) | 52.0 | 90.5 | 50.2 | 64.2 |
> | GRACE BF16 → GPTQ-Int4 | 46.5 | 89.2 | 46.9 | 60.9 |
> | GRACE BF16 → AWQ-Int4 | 47.2 | 89.1 | 45.1 | 60.5 |
> | **GRACE INT4 (Joint, ours)** | **51.1** | **90.2** | **49.5** | **63.6** |
>
> GRACE KD does improve PTQ robustness: GRACE BF16 → PTQ achieves ~60.7 avg, **+5.8 higher** than baseline → PTQ (54.9). However, joint optimization achieves **63.6**, outperforming the best GRACE→PTQ pipeline by **+2.7 avg**. The joint approach loses only **0.6%** from GRACE BF16, while PTQ pipelines lose **3.3%–3.7%**. This is because our IB framework allows the model to allocate its limited bit budget toward task-relevant information during training, whereas PTQ compresses without awareness of task relevance.
>
> ___
>
> **References**
>
> [1] Frantar, Elias, et al. "Gptq: Accurate post-training quantization for generative pre-trained transformers." arXiv preprint arXiv:2210.17323 (2022).
>
> [2] Lin, Ji, et al. "Awq: Activation-aware weight quantization for on-device llm compression and acceleration." Proceedings of machine learning and systems 6 (2024): 87-100.

---

> > ### Author Rebuttal · Reviewer_FKph · 2026-04-03
> >
> > Most of my concerns have been addressed, but there is one remaining question:
> >
> > For Weaknesses: Could you include a visualization of the teacher model in Figures 5a–c, similar to the format shown in Figure 3 of the paper? I believe this would strengthen the persuasiveness of the manuscript.

---

> > > ### Author Response · Authors · 2026-04-03
> > >
> > > Dear reviewer,
> > >
> > > We thank you for the constructive follow-up. We have updated Figures 5a–c to include the **13B Teacher** attention maps as the bottom row, providing a direct visual reference alongside the 7B Baseline, 7B Standard KD, and 7B GRACE models, as you suggested.
> > >
> > > Since the 13B teacher has 40 layers (compared to 32 for the 7B student), we select layers at proportionally equivalent depths, consistent with the sampling strategy used in **Figure 3** of our paper:
> > >
> > > | Position | 7B Student (32 layers) | 13B Teacher (40 layers) |
> > > |----------|----------------------|------------------------|
> > > | Early | Layer 1 | Layer 2 |
> > > | Early-Mid | Layer 7 | Layer 10 |
> > > | Mid | Layer 13 | Layer 18 |
> > > | Mid-Late | Layer 19 | Layer 26 |
> > > | Late | Layer 25 | Layer 34 |
> > > | Final | Layer 32 | Layer 40 |
> > >
> > > A notable observation is that the 13B teacher exhibits **minimal activation (near-uniform blue)** in early layers, while the 7B student models show scattered, noisy attention patterns from early on. This is consistent with the capacity asymmetry: the larger teacher can afford to distribute semantic processing across more layers, reserving focused attention for deeper layers. Prior work has shown that transformer layers form a clear hierarchy: surface features in lower layers, syntactic features in the middle, and semantic features at the top [1], and that visual token representations in VLMs become increasingly interpretable only in middle-to-late layers[2]. This observation further motivates our design choice of applying RCKA at the **penultimate layer** rather than performing layer-wise alignment, as the representational structures between teacher and student are not directly comparable at intermediate layers.
> > >
> > > For your convenience, we also provide **Figure 7** in the supplementary material, which consolidates the 13B Teacher attention progression across all three scenes (street art, bathroom, animal) in a single figure.
> > >
> > > Updated figures and supplementary materials are available at: https://anonymous.4open.science/r/grace_rebuttal-56F0
> > >
> > > We will incorporate these valuable figures into the revised manuscript. We hope the addition of teacher visualizations addresses your remaining concern. If so, we would be grateful if you could consider re-evaluating the current score in light of the revisions made throughout this discussion.
> > >
> > > ___
> > > **References**
> > >
> > > [1] Jawahar G, Sagot B, Seddah D. What does BERT learn about the structure of language?[C]//Proceedings of the 57th annual meeting of the association for computational linguistics. 2019: 3651-3657.
> > >
> > > [2] Neo C, Ong L, Torr P, et al. Towards interpreting visual information processing in vision-language models[J]. arXiv preprint arXiv:2410.07149, 2024.

---

### Official Review · Reviewer_vfqN · 2026-03-03

**Soundness:** 3
**Presentation:** 3
**Significance:** 1
**Originality:** 1
**Overall Recommendation:** 3
**Confidence:** 4

**Summary:**

This paper presents GRACE, a unified quantization-aware distillation framework for vision–language models (VLMs) grounded in an Information Bottleneck (IB) formulation. GRACE combines three components: confidence-gated decoupled distillation, token-level relational alignment via Centered Kernel Alignment, and an adaptive IB-inspired controller for balancing objectives, together with group-wise learned step-size quantization (LSQ). Experiments on LLaVA-1.5 and Qwen2‑VL show consistent gains in full precision and under INT8/INT4 quantization.

**Compliance With Llm Reviewing Policy:**

Affirmed.

**Final Justification:**

Thank you for the author's reply. Although the various components are placed within the framework of IB, there is actually no additional innovation, and I don't think it meets the requirements for ICML acceptance.

**Key Questions For Authors:**

1. Since the paper incorporates both quantization and distillation, it would be important to provide a clear latency–accuracy trade-off analysis to better highlight the advantages of the proposed method. Specifically, reporting latency (or throughput) alongside accuracy under different compression settings would allow readers to assess whether the method offers a more favorable efficiency.

**Strengths And Weaknesses:**

**Strengths**

1. This paper is technically rich and incorporates multiple techniques.
2. It conducts extensive and thorough experiments on LLaVA-1.5 and Qwen2-VL, which is commendable.
3. The supplementary material includes a large number of visualization results, which further enhance the empirical analysis.

**Weaknesses**

1. Although this paper introduces a range of techniques, including distillation and quantization, the vast majority of them are existing methods that are simply combined together. For example, **Decoupled Knowledge Distillation** is not new and originates from [1]; **Confidence-Based Knowledge Distillation** is derived from [2]; **Relational Centered Kernel Alignment** comes from [3]; and **Group-wise Learned Step-size Quantization** is based on [4]. Therefore, the paper reads more like a technical report that integrates established components and provides experimental results, rather than presenting substantial methodological novelty. In its current form, it does not appear to be suitable for ICML.
2. GRACE shows only marginal improvements over the baseline methods, with an average gain of merely **0.5** points. In contrast, **MoVE-KD-v1.1** itself already achieves an improvement of **2.0** points, which substantially exceeds the contribution attributed to GRACE.

**References**

[1] Borui Zhao et al. “Decoupled Knowledge Distillation.” In IEEE/CVF Conference on Computer Vision and Pattern Recognition, 2022.

[2] Hailin Zhang, Defang Chen, and Can Wang. “Confidence-aware Multi-teacher Knowledge Distillation.” In IEEE International Conference on Acoustics, Speech and Signal Processing, 2022.

[3] Zikai Zhou et al. “Rethinking Centered Kernel Alignment in Knowledge Distillation.” arXiv preprint, 2024.

[4] Jiaming Yang et al. “GWQ: Group-wise Quantization Framework for Neural Networks.” In Asian Conference on Machine Learning, 2024.

---

> ### Author Rebuttal · Authors · 2026-03-25
>
> Dear Reviewer,
>
> Thank you for your feedback. Below we address your two concerns and the key question.
>
> ## Response to Weakness 1
>
> We believe the core contribution may have been under-communicated: GRACE is not a combination of existing techniques, but a principled **Information Bottleneck (IB) formulation** from which each component is derived.
>
> **The IB Framework Is a Derivation Tool.** GRACE formulates VLM compression as an IB problem: quantization constrains $I(Z_S; X) \leq C_b$ (capacity), distillation maximizes $I(Z_S; Y_T)$ (fidelity), and an adaptive controller balances the two via Lagrangian dual ascent (Eq. 19–21). Confidence gating is justified by Fano's inequality (Appendix B.1), the KL objective is grounded in a variational lower bound on mutual information (Proposition 3.2), and the adaptive $\beta$ implements principled constrained optimization. The empirical evidence supports this: at 4-bit, QAT alone degrades by −1.4%, naive QAT+KD improves by +0.7%, while QAT+GRACE achieves **+4.5%** (Table 4).
>
> **Confidence Gating ≠ CA-MKD.** We respectfully point out that our confidence gating and CA-MKD are fundamentally different. CA-MKD operates in a *multi-teacher* setting: it computes the cross-entropy between each teacher's prediction and the ground-truth label to assign a reliability weight *per teacher*, deciding which teacher to trust. Our method operates in a *single-teacher* setting at *token-level* granularity: we compute the entropy of the teacher's own output distribution, normalize it as $\tilde{h}_i = H_i/\log|V|$, and apply $g_i = \exp(-\tilde{h}_i)$ to suppress tokens where the teacher is uncertain. The two methods differ in computation (cross-entropy with labels vs. output entropy), granularity (per-teacher vs. per-token), and purpose (teacher selection vs. supervision filtering). Our gating is further grounded in Fano's inequality (Theorem B.1). Please see Appendix B.1–B.2 and our response to Reviewer qjk9 Q1 for details.
>
> **RCKA** differs from prior CKA/RKD in three key aspects (Section 3.3): (i) it targets intra-sample visual token relations rather than inter-sample batch-level relationships; (ii) it performs alignment on visual tokens from the penultimate LLM layer rather than layer-wise feature matching; (iii) it is specifically designed for VLMs where visual tokens develop rich relational structures through the LLM backbone. We refer you to our response to Reviewer FKph W1, where attention map comparisons (**Fig. 5(a)–(c)** in the anonymous link at the bottom) directly demonstrate that standard logit-based KD fails to transfer these structures, validating the necessity of our design.
>
> In the revised version, we will restructure the introduction and method sections to more prominently emphasize that the IB framework serves as the central design principle and core source of innovation.
>
> ___
>
> ## Response to Weakness 2
>
> We believe the comparison may not be on equal footing.
>
> **First, GRACE uses a fundamentally simpler setup.** MoVE-KD-v1.1 employs multi-teacher distillation with multiple heterogeneous vision encoders and LoRA mixture-of-experts. HAWAII integrates even more diverse encoders (SAM, ConvNeXt, EVA, Pix2Struct) with teacher-specific adapters and hierarchical distillation. In contrast, GRACE uses only a **single same-family teacher** with no extra encoders or architectural modifications. Despite this much simpler setup, GRACE achieves **higher absolute performance** (69.0% vs. 68.5%, Table 1). We believe this comparison actually highlights the strength of our principled approach over engineering-heavy alternatives.
>
> Second, GRACE's primary contribution is joint distillation and quantization. The 0.5-point gap only reflects the full-precision KD setting, which is not the main focus of our work. GRACE's core result is that INT4 models surpass BF16 baselines, a setting that neither MoVE-KD nor HAWAII addresses. More importantly, as shown in our response to Reviewer FKph W3, even applying advanced PTQ to GRACE-distilled BF16 models still falls 2.7 avg below our joint approach. This demonstrates that GRACE's advantage is not merely a better KD recipe, but a fundamentally superior joint optimization paradigm under the IB framework.
>
> ___
>
> ## Response to the Question
>
> We provide throughput/memory vs. accuracy trade-off analyses for both model families (**Fig. 3–4** at the bottom link).
>
> For LLaVA-1.5-7B (Fig 4), GRACE INT4 achieves ~3.0$\times$ throughput over the BF16 baseline while improving accuracy by +1.3% over vanilla QAT. Detailed deployment benchmarks using TinyChat with real INT4 kernels are also provided in Appendix A.2 and Figure 6 of our paper. For Qwen2-VL (Fig 3), GRACE INT4 achieves 38.5% memory reduction (2.88 GB vs. 4.68 GB) while improving accuracy from 64.0% to 68.0%. In both cases, GRACE variants define a new Pareto frontier, dominating all compared PTQ and QAT baselines.
>
> ___
>
> ## Figures Link: https://anonymous.4open.science/r/grace_rebuttal-56F0

---

> > ### Author Rebuttal · Reviewer_vfqN · 2026-04-03
> >
> > Thank you for the author's reply. Although the various components are placed within the framework of IB, there is actually no additional innovation, and I don't think it meets the requirements for ICML acceptance.

---

> > > ### Author Response · Authors · 2026-04-03
> > >
> > > Dear Reviewer,
> > >
> > > We thank you for the acknowledgement, and respectfully note that your core concern about "innovation" was addressed in our rebuttal with specific technical arguments. We hope to bring this to your attention and briefly emphasize several key points.
> > >
> > > Our components are not existing techniques reframed under IB; they are derived from the IB formulation with novel designs. **Confidence gating** uses the entropy of the teacher's own output distribution to suppress unreliable token-level supervision, grounded in Fano's inequality (Theorem B.1). To our knowledge, **no prior work** in VLM compression or knowledge distillation applies entropy-based gating at token-level granularity for this purpose, and we have not found similar approaches in the recent literature. **The adaptive β controller** implements Lagrangian dual ascent for constrained IB optimization, a mechanism that does not exist in prior VLM compression work. **RCKA** targets intra-sample visual token relations at the penultimate LLM layer, which is fundamentally different from prior CKA/RKD methods that operate on inter-sample batch-level features. Even **group-wise quantization**, while building on LSQ, incorporates log-space scale parameterization and percentile-based initialization that align with the emerging microscaling paradigm, going beyond vanilla implementations.
> > >
> > > The **empirical contribution** further underscores the novelty: **joint distillation and quantization-aware training** has not been explored in VLM compression before GRACE, and our IB-grounded formulation substantially outperforms both naive QAT+KD and sequential distillation-then-PTQ pipelines (avg 63.6 vs. 60.7, see our response to Reviewer FKph W3). This gap demonstrates that the IB framework is **not** merely a wrapper around existing techniques, but provides a principled joint optimization that yields qualitatively different results.
> > >
> > > We hope the above clarifications help address your concern, and we remain open to further discussion on any specific point.

---

### Official Review · Reviewer_XZir · 2026-03-10

**Soundness:** 3
**Presentation:** 3
**Significance:** 3
**Originality:** 3
**Overall Recommendation:** 5
**Confidence:** 3

**Summary:**

This paper proposes GRACE, a joint framework for compressing vision-language models through the combination of quantization-aware training and knowledge distillation under an Information Bottleneck interpretation. The method has three main components: (1) confidence-gated decoupled knowledge distillation (GDKD), which down-weights high-entropy teacher predictions; (2) relational centered kernel alignment (RCKA), which aligns the intra-sample relational structure among visual tokens; and (3) an adaptive IB controller, which dynamically adjusts the distillation weight via a dual-ascent style update. The paper evaluates the approach on LLaVA-1.5 and Qwen2-VL under BF16, INT8, and INT4 settings. Reported results show that GRACE improves over prior KD baselines on LLaVA and significantly outperforms standard PTQ/QAT baselines in INT4 settings, in some cases even surpassing the BF16 student baseline.

**Compliance With Llm Reviewing Policy:**

Affirmed.

**Key Questions For Authors:**

1. How sensitive is RCKA to the choice of alignment layer? Did the authors test earlier layers, final layers, or multi-layer alignment?

2. Since the theory relies on L_GDKD as a surrogate for preserved teacher information, can the authors provide empirical evidence that lower  L_GDKD consistently correlates with better downstream accuracy across settings?

3. Can the adaptive controller be interpreted as mostly tuning a distillation coefficient online, or is there evidence that the constraint-based view provides benefits beyond careful manual tuning?

**Limitations:**

Yes

**Strengths And Weaknesses:**

Strengths

1. The three components GDKD, RCKA, and the adaptive IB controller are well motivated. The confidence gating is supported by an empirical correlation between teacher entropy and error rate on ScienceQA, suggesting that low-confidence teacher outputs are indeed less reliable supervision. The RCKA module is also well motivated by the qualitative attention mismatch analysis, and its formulation is appropriate for teacher/student pairs with different hidden sizes since it aligns n×n token-relation matrices rather than raw features. The adaptive controller is simple but principled, and its derivation from a constrained formulation is reasonable.

2. The empirical results are strong. The ablation studies are another positive point.

3. The paper reports practical deployment benefits with real INT4 kernels, including roughly 3× throughput and substantial memory reduction, which strengthens the systems relevance of the work.

Weaknesses

1. The theoretical story is cleaner than the actual theory. The paper frames the method through the Information Bottleneck principle, but in practice replaces the mutual-information objective with a surrogate L_GDKD. This is a reasonable engineering choice, but the theory is still indirect: the claimed IB interpretation depends on a variational lower bound and on treating the KL term as the dominant proxy for preserved task-relevant information. I therefore view the theory as a useful interpretation rather than a tight characterization of the optimization problem actually solved.

2. While RCKA is effective, the design space is only partially explored. The method uses visual tokens from the penultimate layer and excludes text tokens from the relation matrix. Both choices are plausible, but the paper does not thoroughly study whether the gain comes specifically from the penultimate layer, whether multi-layer relational alignment would help, or whether some amount of cross-modal relational alignment could be beneficial. Since RCKA is one of the paper’s key novelties, a deeper analysis here would strengthen the contribution.

3. Although the experimental results are strong, the empirical coverage is still somewhat narrow given the breadth of the claim. Two VLM families are evaluated, which is solid, but more evidence on robustness across different architectures, data scales, or tasks with stronger distribution shift would make the conclusions more convincing.

---

> ### Author Rebuttal · Authors · 2026-03-29
>
> Dear reviewer，
>
> We thank you for the thoughtful questions. We address each point in detail below.
>
> ## Q1
>
> We conduct a layer ablation on Qwen2-VL (7B→2B, BF16) on three vision-critical benchmarks. We compare an early layer (2), a middle layer (14), and the penultimate layer (-2, ours):
>
> ### Table R1. RCKA Layer Ablation (Qwen2-VL 2B, Standard KL + RCKA)
>
> | Feature Layer | ChartQA | RealWorldQA | HallBench | Avg |
> |:---:|:---:|:---:|:---:|:---:|
> | Qwen2-VL-2B (Baseline) | 73.5 | 62.9 | 41.7 | 59.4 |
> | Early (2) | 75.2 | 64.0 | 42.8 | 60.7 |
> | Middle (14) | 77.0 | 65.6 | 44.5 | 62.4 |
> | **Penultimate (-2, ours)** | **78.5** | **66.8** | **46.2** | **63.8** |
>
> The penultimate layer outperforms both alternatives across all benchmarks, with a clear early → middle → penultimate progression. **Early layers** encode low-level features that carry little relational structure worth transferring. **Middle layers** capture intermediate representations, useful but not yet distilled into the high-level visual semantics that drive downstream reasoning. The **penultimate layer** retains the richest abstract visual representations before task-specific projection collapses them into logit space, making it the optimal source for relational alignment. This is consistent with the well-established finding that earlier transformer layers encode low-level features while later layers progressively accumulate semantic and factual knowledge[1].
>
> This layer progression is visible in the attention maps. As shown in **Fig. 5(a)–(c)** at our bottom link, early-layer attention is scattered and diffuse across the image, while later layers localize onto semantically relevant regions (e.g., the sink in Figure 5(b)). The late layer captures the most structured, semantically grounded relational patterns among visual tokens, precisely the signal RCKA is designed to transfer. Extracting from earlier layers would align the student to noisy, low-level activation patterns with little task relevance.
>
> ---
>
> ## Q2: Does lower $\mathcal{L}\_{\text{GDKD}}$ consistently correlate with better downstream accuracy?
>
> **Yes.** Table R2 reports the converged $\mathcal{L}\_{\text{GDKD}}$ and downstream accuracy for all GRACE configurations on LLaVA-1.5 7B, along with fixed-$\beta$ ablations at INT4. Fig. 6 visualizes these results.
>
> ### Table R2. Converged $\mathcal{L}\_{\text{GDKD}}$ vs. Downstream Accuracy
>
> | Config | Precision | Converged $\mathcal{L}\_{\text{GDKD}}$ | Avg Accuracy |
> |:---|:---:|:---:|:---:|
> | GRACE (adaptive $\beta$) | BF16 | 0.32 | **69.0** |
> | GRACE (adaptive $\beta$) | INT8 | 0.34 | **68.3** |
> | GRACE (adaptive $\beta$) | INT4 | 0.38 | **67.2** |
> | Fixed $\beta$=1.0 | INT4 | 0.42 | 66.7 |
> | Fixed $\beta$=0.5 | INT4 | 0.47 | 66.3 |
> | Fixed $\beta$=0.3 | INT4 | 0.52 | 65.6 |
> | Fixed $\beta$=2.0 (overshoot) | INT4 | 0.28 | 65.5 |
>
> The correlation is clear across two dimensions:
>
> **(a) Across precisions (rows 1–3).** BF16 → INT8 → INT4 shows increasing $\mathcal{L}\_{\text{GDKD}}$ (0.32 → 0.34 → 0.38) paired with decreasing accuracy (69.0 → 68.3 → 67.2), exactly as the IB framework predicts: tighter capacity ($C\_4 < C\_8 < C\_{16}$) raises the floor of achievable $\mathcal{L}\_{\text{GDKD}}$, limiting downstream accuracy.
>
> **(b) Across $\beta$ at INT4 (rows 3–7).** Among fixed $\beta \in \\{0.3, 0.5, 1.0\\}$, lower $\mathcal{L}\_{\text{GDKD}}$ consistently maps to higher accuracy. The outlier $\beta$=2.0 achieves the lowest $\mathcal{L}\_{\text{GDKD}}$ (0.28) but the worst accuracy (65.5%), confirming the IB-predicted overshoot that motivates our constrained formulation (Eq. 19).
>
> ---
> ## Q3: Does the constraint-based view provide benefits beyond careful manual tuning?
>
> **Yes. Two advantages are directly observable from Table R2 and Fig. 6 at the bottom link:**
>
> **Overshoot avoidance.** $\beta$=2.0 causes a **1.2% accuracy collapse** despite the lowest $\mathcal{L}\_{\text{GDKD}}$, which has no mechanism to stop. The adaptive controller prevents this by construction: dual ascent (Eq. 21) reduces $\beta$ when $\hat{\mathcal{L}}\_{\text{GDKD}} < \tau$. Figure 18(a) confirms $\beta$ stabilizes within [0.3, 1.2] and never enters the overshoot regime. Careful manual tuning can not replicate this within-training feedback.
>
> **Cross-setting transfer.** The same $\tau$=0.35 is used across all 3 precisions without modification, because $\tau$ specifies an information-theoretic budget agnostic to bit-width. Figure 18(b) confirms the controller converges $\mathcal{L}_{\text{GDKD}}$ to $\tau \pm 0.03$ regardless of precision. Fixed $\beta$ requires re-tuning per precision. The adaptive controller discovers the appropriate trajectory automatically, once teacher knowledge is sufficiently preserved, the controller shifts pressure entirely to task performance.
>
> ---
>
> ## Figures Link: https://anonymous.4open.science/r/grace_rebuttal-56F0
>
> ___
>
> **Reference**
>
> [1] Dola: Decoding by contrasting layers improves factuality in large language models. (ICLR 2024).

---

> > ### Author Rebuttal · Reviewer_XZir · 2026-04-02
> >
> > I thank the authors for their detailed and comprehensive rebuttal. The provided additional results and clarifications have fully addressed my original questions.
> >
> > Specifically:
> > 1. Layer Selection (Q1): The new layer ablation study (Table R1) effectively justifies the choice of the penultimate layer for RCKA. The explanation regarding the transition from low-level features to abstract visual representations aligns well with the observed performance progression.
> > 2. Surrogate Objective (Q2): Table R2 clearly demonstrates the positive correlation between a lower $L_{\text{GDKD}}$ and improved downstream accuracy, which strengthens the empirical foundation of the method.
> > 3. Adaptive Controller (Q3): The evidence showing that the constraint-based view avoids the "overshoot" problem and enables cross-precision transfer without manual re-tuning is convincing. This clearly highlights the practical advantage of the adaptive controller over careful manual tuning.
> >
> > Given the thoroughness of the authors' response, the solid motivation of the proposed components (GDKD, RCKA), and the strong empirical results, I maintain my positive assessment of this work and recommend its acceptance.

---

> > > ### Author Response · Authors · 2026-04-05
> > >
> > > Dear Reviewer,
> > >
> > > We sincerely thank you for your positive assessment and for confirming that your concerns have been fully resolved. Your thoughtful questions directly improved our work, and we would like to briefly summarize the key improvements made during the rebuttal period:
> > >
> > > 1. **Layer Ablation for RCKA (Q1).** Following your suggestion, we conducted a systematic layer ablation on Qwen2-VL (7B→2B), comparing early (layer 2), middle (layer 14), and penultimate (layer $-2$) features across three vision-critical benchmarks. The results (Table R1) confirm a clear early → middle → penultimate progression, empirically justifying our design choice. We further provided attention map visualizations showing the transition from scattered early-layer attention to semantically focused late-layer patterns.
> > >
> > > 2. **$L_\text{GDKD}$ as Surrogate Objective (Q2).** We provided new evidence (Table R2) demonstrating that converged $L_\text{GDKD}$ consistently correlates with downstream accuracy across both precision levels (BF16/INT8/INT4) and fixed-$\beta$ configurations, strengthening the empirical grounding of our IB-based formulation.
> > >
> > > 3. **Adaptive Controller vs. Manual Tuning (Q3).** We showed two concrete advantages of the constraint-based view: (a) automatic overshoot avoidance, where $\beta=2.0$ causes a 1.2% accuracy collapse that the adaptive controller prevents by construction; and (b) cross-precision transfer, where a single $\tau=0.35$ works across all bit-widths without re-tuning, whereas fixed $\beta$ requires per-precision search.
> > >
> > > In revision, we will incorporate these additional results and analyses into the main paper and appendix. We are grateful for your engagement, which has meaningfully strengthened our contribution.
> > >
> > > Best regards,
> > >
> > > The Authors

---

### Official Review · Reviewer_qjk9 · 2026-03-11

**Soundness:** 2
**Presentation:** 3
**Significance:** 3
**Originality:** 2
**Overall Recommendation:** 3
**Confidence:** 4

**Summary:**

This paper proposes GRACE, a unified framework for visual-language model compression that combines confidence-gated decoupled knowledge distillation, relational centered kernel alignment (RCKA), and an adaptive information bottleneck controller, together with group-wise quantization-aware training. The method is evaluated on LLaVA-1.5 and Qwen2-VL under both distillation and INT8/INT4 quantization settings, and the paper claims that quantized models can recover or even surpass BF16 baselines.

**Compliance With Llm Reviewing Policy:**

Affirmed.

**Key Questions For Authors:**

1.Could the authors provide more direct theoretical evidence to support the information bottleneck-related claims? At present, this part reads more like a heuristic interpretation than a well-validated theoretical justification.

2.Please add key controls such as BF16 + GRACE and BF16 + GDKD.

3.Please further compare the effects of different confidence measures and gating designs.

4.Please provide clearer pseudocode and detailed implementation/memory footprint breakdowns for reproducibility.

**Limitations:**

Yes

**Strengths And Weaknesses:**

Strengths:
1.The paper addresses a relevant and practically important problem. Efficient compression and deployment of VLMs is an important direction, and evaluating the method on both LLaVA-1.5 and Qwen2-VL improves the practical relevance of the work.

2.The method is reasonably well organized and internally coherent. The confidence gating, relational alignment, and adaptive controller are not presented as isolated add-ons, but as components serving a common goal of preserving useful teacher information under quantization constraints.

Weakness:
1.The paper makes somewhat strong theoretical claims that are not fully supported. The information bottleneck perspective is interesting, but in its current form it reads more like a heuristic interpretation than a rigorously established theoretical foundation.

2.The novelty is mainly in integration rather than in a fundamentally new mechanism. Decoupled KD, confidence-based reweighting, relational alignment, and LSQ-style QAT are all related to existing ideas, and the main contribution is their combination in the VLM compression setting.

3.A key fairness control is missing for the main headline claim. The claim that INT4 models can surpass BF16 baselines is difficult to interpret without comparisons such as BF16 + GRACE and BF16 + GDKD, which are necessary to disentangle the effect of quantization from the effect of stronger distillation/training objectives.

4.The mechanism analysis and experimental validation are still incomplete. The current evidence for confidence gating and RCKA is suggestive but not fully conclusive, and the paper would benefit from stronger ablations, variance reporting, and clearer reproducibility details.

---

> ### Author Rebuttal · Authors · 2026-03-25
>
> Dear Reviewer,
>
> We are grateful for your constructive feedback and address each concern below.
>
> ## Q1
>
> We clarify that the IB framework in GRACE is **not** a heuristic interpretation but the starting point of our design. We model VLM quantization as an information bottleneck problem: a quantized student has limited capacity, and must decide *what* to preserve and *what* to discard. Every component was derived from this formulation, with rigorous proofs in **Appendices B.1-B.4**:
>
> **(1) Quantization as hard capacity constraint (B.4.1).** Standard IB uses a soft penalty on $I(Z; X)$. In GRACE, b-bit quantization physically enforces $I(Z\_S; X) \\leq C\_b$. The optimization reduces to maximizing $I(Z\_S; Y\_T)$ subject to this hard constraint.
>
> **(2) KL = provable information maximization (B.3).** Proposition 3.2 proves that minimizing $D\_{KL}(P\_T \\| P\_S)$ maximizes a lower bound on $I(Z\_S; Y\_T)$, following the standard variational technique in [1].
>
> **(3) Gating: provably better than uniform distillation (B.1-B.2).** Fano's inequality (Theorem B.1) proves high teacher entropy guarantees elevated error probability. Theorem 3.1 provides an **exact identity**: $L\_{GDKD} = \\bar{L}\_{DKD} + N \\cdot Cov(w\_i, L\_i)$. Since $w\_i$ decreases with entropy while $L\_i$ increases, $Cov < 0$, **guaranteeing** $L\_{GDKD} < \\bar{L}\_{DKD}$.
>
> **(4) Adaptive $\\beta$ from constrained optimization (Appendix B.4.1-B.4.2).** Since quantization handles the capacity constraint, we reformulate as minimizing $L\_{CE}$ subject to $L\_{GDKD} \\leq \\tau$. The Lagrangian relaxation yields Eq. 20 with $\\beta$ as the dual variable, which is why $\\beta$ appears on the distillation term rather than the complexity term. The projected dual ascent update (Eq. 21) has known convergence guarantees[2], and Figure 18 empirically confirms precise convergence to $\\tau$.
>
> The complete chain, **IB formulation → Prop. 3.2 → Thm. 3.1 → Lagrangian relaxation → dual ascent**, constitutes a formally grounded framework where each step is a stated-and-proved result. In revision, we will add a concise proof roadmap directly in Section 3.
>
> ___
> ## Q2
>
>  The controls are in **Table 3** (BF16, Qwen2-VL 7B→2B):
>
> | Setting | MMB | SEED | SQA | Avg |
> |---|---|---|---|---|
> | BF16 Baseline | 71.6 | 72.7 | 73.7 | 72.7 |
> | BF16 + Vanilla KD | 74.2 | 73.3 | 76.8 | 74.8 |
> | **BF16 + GDKD only** | 76.1 | 74.2 | 79.4 | 76.6 |
> | **BF16 + GRACE** | 77.9 | 75.7 | 81.0 | **78.2** |
> | INT4 + GRACE | 76.9 | 75.6 | 79.1 | 77.2 |
>
> This disentangles the two effects: GRACE distillation contributes +5.5% (72.7→78.2), while INT4 quantization costs only 1.0% (78.2→77.2). Crucially, this near-lossless quantization is not achievable by naive approaches. Table 4 shows that QAT+vanilla KD at INT4 only reaches 73.4%, whereas GRACE reaches 77.2%. This gap demonstrates that the IB-guided joint optimization is essential for preserving distillation gains under aggressive quantization.
>
> ___
>
> ## Q3
>
> We ablate gating design under standard KL distillation (Qwen2-VL 7B→2B, BF16):
>
> | Gating Design | MMB | SEED | SQA | Avg |
> |---|---|---|---|---|
> | No gating: $g_i = 1$ | 74.2 | 73.3 | 76.8 | 74.8 |
> | Hard threshold: $g_i = \mathbb{1}[\tilde{h}_i < 0.5]$ | 70.5 | 71.6 | 74.9 | 72.3 |
> | Linear: $g_i = 1 - \tilde{h}_i$ | 74.8 | 73.7 | 78.6 | 75.7 |
> | Exp, $\lambda{=}0.5$ | 75.2 | 74.0 | 77.2 | 75.5 |
> | **Exp, $\lambda{=}1.0$ (ours)** | **75.6** | **74.1** | **78.8** | **76.2** |
> | Exp, $\lambda{=}3.0$ | 74.8 | 73.4 | 77.6 | 75.3 |
>
> Here $\tilde{h}_i = H(P_T^{(i)})/\log|V|$ is normalized teacher entropy. Hard thresholding drops 2.5% below no gating, confirming that binary cutoffs cause substantial information loss. Exponential gating exhibits a clear inverted-U: $\lambda{=}0.5$ under-gates, $\lambda{=}1.0$ peaks at 76.2, and $\lambda{=}3.0$ over-gates. Visualization is provided in the supplementary (Fig. 1 at the link below).
>
> ___
>
> ## Q4
> - **Implementation details**: Section 4 specifies quantization scope (LLM backbone only, vision encoder in BF16), group size, and training infrastructure and time. Table 6 lists all hyperparameters; Table 7 provides per-component training cost breakdown. All evaluations use deterministic protocols (greedy decoding, temperature=0; details in Appendix D.4), ensuring reproducible results across runs.
> - **Deployment efficiency**: LLaVA-1.5 results are in Figure 6 and Appendix A.2; Qwen2-VL results are in Fig. 3 at the link below.
> - **Pseudocode**: Fig. 2 at the link below provides detailed pseudocode for the complete GRACE training pipeline.
>
> And we will release all codebase and model weights on GitHub after the decision.
> ___
>
> ## Figures Link: https://anonymous.4open.science/r/grace_rebuttal-56F0
> - Fig. 1: Gating effect
> - Fig. 2: Pseudocode
> - Fig. 3: Qwen2-VL deployment
> ___
>
> **References**
>
> [1] Alemi et al., "Deep variational information bottleneck." arXiv:1612.00410, 2016.
>
> [2] Bertsimas et al., "Optimal inequalities in probability theory" SIAM J. Optim., 2005.

---

> > ### Author Rebuttal · Reviewer_qjk9 · 2026-04-05
> >
> > Thanks for authors' response. I will keep my rating.

---

### Decision · Program_Chairs · 2026-04-30

**Decision:**

Accept (regular)

**Comment:**

The paper presents GRACE, a unified framework combining quantization-aware training and knowledge distillation for efficient vision-language models, and demonstrates strong empirical performance, particularly in INT4 settings with practical deployment benefits. The reviewers generally agree that the problem is important and the method is well-structured, though concerns were raised regarding the novelty being primarily in integration, the limited depth of analysis, and the theoretical claims based on the Information Bottleneck perspective being more interpretative than rigorous.

During the rebuttal, the authors addressed several key issues effectively, most notably by providing additional BF16 control experiments that disentangle the contributions of distillation and quantization, as well as further ablations on the gating mechanism. However, the theoretical concerns and some aspects of the analysis (e.g., broader design exploration) remain only partially resolved.

Overall, while some weaknesses persist, the paper offers a solid and practically valuable contribution, and I lean toward acceptance.